# GE-FM: Geometry-aware Energy-based Flow Matching for Non-Euclidean Manifolds

**Ayush Roy**[*]                                                                                    *aroy25@buffalo.edu*
*Department of Computer Science and Engineering*
*University at Buffalo, SUNY*

**Arjun Ramesh Kaushik***                                                                           *kaushik3@buffalo.edu*
*Department of Computer Science and Engineering*
*University at Buffalo, SUNY*

**Vishnu Suresh Lokhande**                                                                          *vishnulo@buffalo.edu*
*Department of Computer Science and Engineering*
*University at Buffalo, SUNY*

**Nalini K Ratha**                                                                                  *nratha@buffalo.edu*
*Department of Computer Science and Engineering*
*University at Buffalo, SUNY*

**Venu Govindaraju**                                                                                *govind@buffalo.edu*
*Department of Computer Science and Engineering*
*University at Buffalo, SUNY*

**Reviewed on OpenReview:** *https://openreview.net/forum?id=Wb9wwUF7aV*

## Abstract

Flow Matching has emerged as a powerful framework for generative transport and denoising, yet existing formulations are inherently Euclidean, neglecting the curved and time-evolving geometry of diffusion manifolds. Recent higher-order extensions seek to recover curved transport by explicitly modeling higher derivatives, but these approaches introduce instability and accumulate discretization error, particularly in few-step ODE sampling regimes. We propose a strictly first-order-in-time, energy-based flow matching framework that incorporates geometry through Christoffel-adjusted dynamics. Our method defines a total energy as the sum of kinetic energy induced by the predicted velocity field and a learned potential energy, and enforces approximate energy conservation along transport trajectories. Energy conservation encourages optimal low-energy denoising paths and yields a smoother optimization landscape, leading to faster and more stable convergence. Crucially, the energy formulation induces a time-dependent Riemannian metric that captures the evolving diffusion geometry without explicit manifold supervision. Christoffel symbols derived from this induced metric adjust the velocity field to account for curvature, implicitly modeling higher-order effects without introducing additional learnable dynamics. This geometric correction is manifold-agnostic and adapts automatically to the evolving diffusion structure. Empirically, our method outperforms existing few-step baselines, achieving improved performance on both FID and mode coverage ($\sim 80\% \uparrow$ on synthetic spiral datasets). To the best of our knowledge, this is the *first geometry-aware flow matching framework that integrates energy conservation and Christoffel dynamics for stable curved generative transport*. The code used in this work is available at: `https://github.com/AyushRoy2001/GE-FM.git`.

---

[*]Equal contributions.

# 1 Introduction

Diffusion models have become the de facto generative framework in modern machine learning, achieving state-of-the-art performance in high-fidelity visual generation and understanding tasks Jiang et al. (2025); Lukoianov et al. (2024); Lipman et al. (2022). Despite their impressive performance, practical deployment remains challenging due to high inference latency, as generating a sample typically requires numerous sequential denoising steps. Few-step diffusion has recently emerged as a promising alternative, but this acceleration often comes at the cost of degraded generation quality (see Section A).

This limitation motivated the development of a new class of diffusion models, *Flow Matching* Lipman et al. (2023), which learn continuous transport maps between distributions by encouraging straighter trajectories, thereby enabling sampling with significantly fewer steps. While flow matching improves efficiency, a substantial performance gap between few-step and multi-step diffusion models remains. MeanFlow Geng et al. (2025) narrows this gap by stabilizing training and improving the integration of classifier-free guidance. Building on this, AlphaFlow Zhang et al. (2025) identifies competing gradient components in MeanFlow and proposes a generalized optimization objective that enforces velocity consistency along the trajectory. Despite these advances, the performance of few-step generation remains limited due to the curvature bias introduced with highly curved manifolds.

We argue that this limitation is not merely *an optimization issue*, but a consequence of deeper modeling assumptions in existing flow matching frameworks. These methods parameterize transport through velocity fields whose outputs are composed additively. The update $x_{t+d} = x_t + d \cdot v(x_t, t, d)$ treats each step as locally linear regardless of the step size $d$. This composition rule is exact only when the velocity field is constant along the interval $[t, t + d]$. For small $d$, the approximation is benign; the accumulated error is $\mathcal{O}(d^2)$ per step and can be reduced by taking more steps. But, in the few-step regime, $d$ is significantly large. Each step must transport samples across a significant fraction of the trajectory, traversing regions where the velocity field may rotate, accelerate, or change magnitude.

The self-consistency training objective used by shortcut models attempts to compensate for this: the velocity $v(x_t, t, d)$ is trained to match the average of two half-step predictions, $\frac{1}{2}[v(x_t, t, d/2) + v(x_{t+d/2}, t+d/2, d/2)]$. This midpoint composition implicitly captures acceleration through the velocity difference between two half-steps. However, the target itself is trained under an $\ell_2$ loss, which has a well-known mode-averaging failure: when the optimal large-step velocity is multi-valued (different source samples at the same $(x_t, t)$ should be transported in different directions), the $\ell_2$ minimizer is the conditional expectation (a single vector that averages over all plausible transport directions) Bertsekas (2014). In the few-step regime, this averaging is most severe precisely where it matters most: at early times when the source distribution is broad, and the velocity field must simultaneously resolve multiple target modes.

The result is a fundamental tension. Reducing the number of steps increases $d$, which demands that each velocity prediction encode more global information about the trajectory. But the combination of additive composition and $\ell_2$ averaging systematically discards this information: *the curvature, acceleration, path-length variation and other higher-order dynamics that distinguish one trajectory from another*. What remains is a blurred consensus velocity that transports mass to the right region on average, but fails to preserve the fine-grained structure of the target distribution. A natural response to the limitations of linear composition is to model higher-order dynamics explicitly, predicting not just the velocity $v$ but also its time derivative $\frac{dv}{dt}$ or second derivative $\frac{d^2v}{dt^2}$, and constructing higher-order integrators accordingly. We show empirically that this approach is ineffective in the few-step regime: the Taylor expansion $x_{t+d} \approx x_t + d \cdot v + \frac{d^2}{2}(\frac{dv}{dt}) + \cdots$ diverges rapidly for large $d$ on highly curved transport paths, and training networks to predict these higher derivatives introduces compounding approximation errors that negate the theoretical benefit (Section 4.1).

Inspired by Lagrangian mechanics, rather than supervising the trajectory pointwise through higher-order terms, we impose a *global structural constraint*: **energy conservation along the transport path**. We decompose the total energy into a kinetic term $\frac{1}{2}\|v\|_g^2$, defined with respect to a learned Riemannian metric $g$, and a scalar potential energy $U(x_t, t)$ predicted by an auxiliary network. The conservation constraint $E(t) \approx E(t+d)$ acts as a holonomic constraint on the velocity field, penalizing trajectories that spuriously accelerate or decelerate - the dominant failure mode when large-step predictions average over multiple

transport directions. The energy formulation naturally induces a Riemannian structure on the transport space. The metric $g$, derived from the velocity field's local sensitivity, defines a notion of distance that adapts to the evolving geometry of the diffusion process. This provides the opportunity for the second contribution: we augment the velocity field with **Christoffel symbol corrections** $\Gamma_{ij}^k v^i v^j$ Do Carmo (2016); Lee (2003) that account for the curvature of the transport manifold under $g$. These corrections act as a geometric regularizer, biasing the learned trajectories toward geodesics of the induced metric, which are paths of minimal energy that are maximally compatible with large-step integration. Together, the energy constraint and the geodesic correction form a unified framework: the former controls *what* is conserved along trajectories, while the latter controls *how* trajectories shape through the learned geometry.

We refer to this new family of flow-matching diffusion models as **GE-FM** (Geometric Energy Flow Matching). Through extensive experiments on highly curved synthetic datasets Section 4.1, we empirically validate the proposed framework and isolate the contributions of each modeling component. Furthermore, we validate the effectiveness and the superiority of the proposed method compared to the existing flow matching models for multiple highly non-Euclidean synthetic datasets Section 4.

## 2 Background and Problem Formulation

In this section, we formalize two key limitations of existing flow matching frameworks that motivate our approach. Standard formulations learn first-order velocity fields in ambient Euclidean space, implicitly restricting transport to straight-line geometries (Section A.2). This leads to two complementary failure modes: (i) *statistical fragility of higher-order supervision*, where incorporating acceleration introduces estimation noise that scales poorly with temporal resolution, and (ii) *geometric bias of Euclidean transport*, where straight-line interpolation deviates from intrinsic manifold geodesics on curved data. Together, these reveal a core tension: accurately modeling curved transport requires higher-order geometry, yet explicit higher-order supervision is unstable. We formalize these effects in Lemma 1 and Lemma 2.

### 2.1 Variations of Flow Matching

Flow matching Lipman et al. (2023) casts generative modeling as learning a continuous-time transport between a base distribution and the data distribution. Let $x_0 \sim p_{\mathrm{data}}$, $x_1 \sim p_{\mathrm{noise}}$, and let the path $x(t) \in \mathbb{R}^d$ be defined by $x(t) = \psi(x_0, x_1, t)$, where $t \in [0, 1]$. This induces a ground-truth velocity field $v^*(x(t), t) = \frac{\partial}{\partial t}\psi(x_0, x_1, t)$. A parametric velocity field $v_\theta(x(t), t)$ is learned via

$$\mathcal{L}_{\mathrm{FM}}(\theta) = \mathbb{E}_{x_0, x_1, t}\left[\|v_\theta(x(t), t) - v^*(x(t), t)\|^2\right] \tag{1}$$

The minimizer of equation 1 satisfies

$$v_\theta(x(t), t) = \mathbb{E}\left[v^*(x(t), t) \mid x(t) = x\right] \tag{2}$$

So the learned field equals the conditional mean velocity at each space-time location. Different path choices $\psi(\cdot)$ yield different flow families.

**Mean Flow Geng et al. (2025).** Using linear interpolation, $x(t) = (1-t)x_0 + tx_1$, gives constant per-sample velocity $v^*(x(t), t) = x_1 - x_0$ and learned field $v_{\mathrm{mean}}(x, t) = \mathbb{E}[x_1 - x_0 \mid x(t) = x]$. Mean flows are simple but often strongly time-dependent, producing trajectories and rigid ODE dynamics that slow numerical sampling due to the inability to adapt to the curvature of manifolds.

**Rectified Flow Liu et al. (2022).** Rectified flow reparameterizes the path to align transport directions, typically $x(t) = (1-t)x_1 + tx_0$, with $v^*(x(t), t) = x_0 - x_1$. The learned field is approximately time-independent, yielding near-straight trajectories that reduce stiffness and enable few-step generation. However, rectification remains first-order and does not explicitly regulate higher-order trajectory geometry.

**Shortcut Frans et al. (2024) and Consistency Models Song et al. (2023).** Consistency models learn a trajectory-invariant mapping $f_\theta(x_t, t)$, collapsing the entire ODE path into a single evaluation for true one-step generation. Shortcut models instead learn a step-size-conditioned velocity field $v_\theta(x_t, t, d)$, ensuring a step of size $d$ matches two steps of $d/2$, enabling flexible few- to many-step generation while retaining an

iterative structure. However, neither imposes structural constraints on how the velocity field evolves along the trajectory.

In the subsequent section, we challenge two fundamental assumptions underlying existing flow matching models: (1) that transport dynamics are assumed to be linear and additive, and (2) that transport trajectories follow straight Euclidean paths. By critically examining these assumptions, we motivate a manifold-aware energy-based flow matching technique that serves as a unifying solution to both limitations.

## 2.2 Effect of higher-order terms

As established in Section 2.1, existing flow matching methods parameterize transport entirely through a first-order velocity field $v_\theta(x_t, t)$. The trajectory is recovered by integrating this field forward: $x_{t+d} = x_t + d \cdot v_\theta(x_t, t)$, an update that is exact only when the velocity is constant over $[t, t+d]$. In practice, optimal transport paths between complex distributions are far from straight - they bend, accelerate, and vary in speed as probability mass navigates around regions of low density and converges toward concentrated target modes. All of this structure is invisible to a first-order parameterization: the velocity at a single point encodes no information about how that velocity will change downstream.

The consequences are benign when many integration steps are available, as the local linearization error $\mathcal{O}(d^2)$ per step remains small. But in the few-step regime, each step spans a large fraction of the trajectory, and the linearization must approximate highly nonlinear transport with a single vector. The model compensates by learning a velocity that is, in effect, a weighted average over the curved path it must approximate - *a compromise that systematically underestimates curvature, smooths over acceleration, and produces trajectories that cut through low-density regions rather than following the natural geometry of the transport.* We therefore seek to extend flow matching beyond first-order dynamics. A natural starting point is to explicitly model higher-order temporal derivatives of the transport trajectory (such as acceleration and jerk), which directly characterize how the velocity field curves and twists along the path. Given a transport path $x(t) = \psi(x_0, x_1, t)$, consider the first-order velocity $\dot{x}(t) = \frac{d}{dt}x(t)$, and its higher-order derivatives as $\frac{d^k x(t)}{dt^k}$, where $k \geq 2$.

For each order $k$, a neural network $f_\theta^{(k)}(x, t)$ is trained to approximate the corresponding ground-truth derivative

$$\underbrace{f^{(k)}(x(t), t)}_{\text{ground-truth derivative}} = \frac{\partial^k}{\partial t^k} \underbrace{\psi(x_0, x_1, t)}_{\text{transport path}} \tag{3}$$

using a squared error objective

$$\mathcal{L}_{\text{higher}}^{(k)}(\theta) = \mathbb{E}\left[\|f_\theta^{(k)}(x(t), t) - f^{(k)}(x(t), t)\|^2\right] \tag{4}$$

**Statistical challenges and overfitting.** Higher-order flow matching explicitly supervises temporal derivatives beyond the velocity field to encode curvature and acceleration of transport paths. While first-order supervision aggregates relatively global endpoint information, higher-order targets require accurately resolving local variations of the trajectory $x(t) = \psi(x_0, x_1, t)$ in time. Consequently, the $k$-th order targets in Equation (3) become increasingly sensitive to small perturbations in $(x_0, x_1, t)$, and their empirical estimation is substantially more sample-intensive than that of first-order velocities. For a fixed dataset size and model capacity, this induces a rapidly growing effective noise level in higher-order labels as $k$ increases (Lemma 1), which in turn exacerbates overfitting to idiosyncratic training paths rather than promoting globally coherent transport geometry. This difficulty is amplified in the few-step inference regime. Large integration steps implicitly assume that the learned dynamics are stable and predictive over extended temporal intervals. When higher-order fields are only weakly identified from limited data, small estimation errors in acceleration and higher derivatives can accumulate nonlinearly across these large steps, leading to unstable trajectories and degraded sample quality.

**Assumption 1.** *Let $x(t) \in \mathbb{R}^d$ denote a ground-truth transport path on $[0, 1]$ satisfying $\frac{d^2 x(t)}{dt^2} = a^*(x(t), t)$, $t \in [0, 1]$, with boundary conditions $x(0) = x_0$ and $x(1) = x_1$. Let the true velocity be $v^*(x, t) = \frac{dx(t)}{dt}$. Assume that $a^*$ is uniformly bounded and Lipschitz continuous in $x$, uniformly over $t \in [0, 1]$. All expectations are taken with respect to the joint data distribution over $(x_0, x_1, t)$.*

Under Assumption 1, we can formalize the statistical gap between velocity and acceleration estimation as in Lemma 1 (proof deferred to Section G.1):

---

**Lemma 1** (***Velocity vs. acceleration estimation rates***). *Let $\hat{v}$ and $\hat{a}$ be empirical estimators of the true velocity $v^*(x,t)$ and acceleration $a^*(x,t)$ obtained from $n$ i.i.d. training samples $\{(x_0^{(i)}, x_1^{(i)}, t^{(i)})\}_{i=1}^n$. Assume that accelerations are estimated via finite temporal differences with step size $h > 0$:*

$$\hat{a}(x,t) \approx \frac{\hat{v}(x, t+h) - \hat{v}(x, t-h)}{2h}$$

*Then, under bounded-moment assumptions,*

$$\mathbb{E}\big\|\hat{v}(x,t) - v^*(x,t)\big\|^2 = O(n^{-1}) \implies \mathbb{E}\big\|\hat{a}(x,t) - a^*(x,t)\big\|^2 = O(n^{-1}h^{-4}) \tag{5}$$

---

In practice, $h \ll 1$ is required to accurately resolve local temporal structure for $t \in [0,1]$; the factor $h^{-4}$ dominates and the bound in Lemma 1 translates to Equation (6), highlighting that higher-order supervision is intrinsically much noisier than first-order supervision at fixed sample size.

$$\varepsilon_2 \equiv \mathbb{E}\big\|\hat{a} - a^*\big\| \gg \varepsilon_1 \equiv \mathbb{E}\big\|\hat{v} - v^*\big\| \tag{6}$$

Lemma 1 shows that estimating higher-order derivatives individually is statistically noisy. However, when multiple derivative orders are learned jointly, the additional higher-order constraints act as a form of temporal regularization that can reduce the effective estimation error of lower-order dynamics, as seen in Corollary 1 (see Section G.2 for proof). Although individual higher-order targets are noisy (Lemma 1), learning multiple derivative orders jointly imposes smoothness constraints on the learned trajectory. Higher-order supervision restricts the model to functions whose temporal evolution is consistent across derivatives, effectively reducing the hypothesis space. This acts as a form of temporal regularization: while each higher-order estimate is noisy in isolation, their joint supervision stabilizes the lower-order dynamics by preventing high-frequency or inconsistent solutions. Consequently, introducing additional higher-order terms can mitigate the statistical instability predicted by Lemma 1, provided their contributions are appropriately weighted.

---

**Corollary 1** (**Stabilization via joint higher-order supervision**). *Let $v^*(x,t)$ denote the true velocity and let $a^{*(k)}(x,t)$, $k \geq 2$, denote higher-order temporal derivatives, where $a^{*(k)}(x,t) = \partial_t^{k-1} v^*(x,t)$. Suppose a model jointly estimates $\{\hat{v}, \hat{a}^{(2)}, \ldots, \hat{a}^{(K)}\}$ by minimizing $\mathcal{L}(\theta) = \mathbb{E}\Big[\|\hat{v}_\theta(X,T) - v^*(X,T)\|^2 + \sum_{k=2}^K \lambda_k \|\hat{a}_\theta^{(k)}(X,T) - a^{*(k)}(X,T)\|^2\Big]$, with weights $\lambda_k \geq 0$. Assume:*
*(i) (**bounded higher derivatives**) $\|a^{*(k)}(x,t)\| \leq C$ for all $2 \leq k \leq K$ and all $(x,t)$;*
*(ii) (**finite-difference linkage**) for each $k \geq 2$, the model's $k$-th head is implemented as a $(k-1)$-th order central finite difference of the velocity head with step size $h$, i.e., $\hat{a}_\theta^{(k)}(x,t) = D_h^{k-1} \hat{v}_\theta(x,t)$, $a^{*(k)}(x,t) = \partial_t^{k-1} v^*(x,t)$, where $D_h$ denotes the (central) first-difference operator;*
*(iii) (**i.i.d. sampling**) training samples are i.i.d. and estimation noise across samples is independent. Then the velocity estimator satisfies the bound*

$$\mathbb{E}\|\hat{v} - v^*\|^2 \leq O(n^{-1}) + O\left(\sum_{k=2}^K \lambda_k h^{2(k-1)}\right) \tag{7}$$

*In particular, choosing $\lambda_k$ to decay sufficiently fast makes the additional term small for $h \ll 1$, and increasing $K$ stabilizes $\hat{v}$ by discouraging rapid temporal oscillations. The same conclusion holds when $\hat{a}^{(k)}$ are produced by separate network heads, provided an additional consistency penalty $\sum_{k=2}^K \mu_k \|\hat{a}_\theta^{(k)} - D_h^{k-1} \hat{v}_\theta\|^2$ is included with $\mu_k$ large enough.*

---

**Computational and Optimization Cost.** Beyond statistical considerations, higher-order flow matching introduces substantial computational overhead. Modeling $k$-th order dynamics requires predicting vector fields

in $\mathbb{R}^d$ for each derivative order, increasing both parameter count and memory footprint. Moreover, training objectives involve evaluating and backpropagating through multiple derivative-matching losses, leading to per-iteration complexity that scales approximately linearly with the number of modeled orders. Furthermore, as seen in Corollary 1, estimating higher-order terms stably needs the introduction of more higher-order terms. This is not a practical solution, and alternatives to achieve the advantages of the higher-order terms without explicitly modeling them are needed.

## 2.3 Geometric bias of Euclidean flow matching.

Lemma 1 shows that higher-order supervision is statistically much noisier than first-order velocity learning, particularly when temporal resolution $h \ll 1$ is required to resolve local dynamics. We now show that, even if we ignore statistical noise and assume perfect access to the conditional mean velocity, standard Euclidean flow matching architectures (Section 2.1) incur an irreducible *geometric* bias on highly curved manifolds.

To formalize this, let $\mathcal{M} \subset \mathbb{R}^d$ denote a smooth 1-dimensional data manifold (e.g., a multi-turn spiral) equipped with its intrinsic arclength metric $d_{\mathcal{M}}(\cdot, \cdot)$, and let $d_{\text{Euc}}(x, y) = \|x - y\|_2$ be the ambient Euclidean distance. We consider transport paths of the form

$$x_{\text{lin}}(t) = (1 - t)x_0 + tx_1, \qquad t \in [0, 1], \tag{8}$$

that underlie MeanFlow Geng et al. (2025), rectified flow Liu et al. (2022), and shortcut models Frans et al. (2024), and compare them with intrinsic geodesic paths $x_{\text{geo}}(\cdot)$ on $\mathcal{M}$.

**Assumption 2.** *The data manifold $\mathcal{M} \subset \mathbb{R}^d$ is a smooth embedded curve with bounded curvature $\kappa(x) \in [\kappa_{\min}, \kappa_{\max}]$ for all $x \in \mathcal{M}$, with $\kappa_{\min} > 0$ representing a strictly curved regime. The source and target distributions $\pi_0, \pi_1$ are supported on $\mathcal{M}$.*

In particular, for spirals used in our synthetic benchmarks, $\kappa_{\min}$ is strictly positive and increases with the number of turns.

**Lemma 2** (**Curvature-induced bias of Euclidean transport**). *Let Assumption 2 hold and let $x_0, x_1 \in \mathcal{M}$ with intrinsic distance $d_{\mathcal{M}}(x_0, x_1) = L$. Let $x_{\text{geo}} : [0, 1] \to \mathcal{M}$ denote the constant-speed geodesic between them, and let $x_{\text{lin}}(t) = (1 - t)x_0 + tx_1$ be the Euclidean interpolation. Assume $L$ is sufficiently small so that the chord lies in a tubular neighborhood of the minimizing geodesic segment and the nearest-point projection onto $\mathcal{M}$ is unique on this segment. Then there exists a universal constant $c > 0$ such that for all $t \in [0, 1]$,*

$$d_{\text{Euc}}\big(x_{\text{lin}}(t), \mathcal{M}\big) \geq c \, \kappa_{\min} \, t(1 - t) \, L^2 \tag{9}$$

*Moreover, let $\{t_k\}_{k=0}^N$ be a uniform grid with step $\Delta t = 1/N$, and let $x_{t_k}^{FM}$ be produced by an explicit Euler discretization of the flow-matching ODE $\frac{dx(t)}{dt} = v^*(x(t), t)$ (in ambient coordinates), initialized at $x_0^{FM} = x_0$. Then there exists a constant $C > 0$ such that*

$$\max_{0 \leq k \leq N} \big\|x_{t_k}^{FM} - x_{\text{geo}}(t_k)\big\|_2 \leq C \, \kappa_{\max} \, L^2 \, \Delta t \tag{10}$$

*Moreover, for a time discretization with step size $\Delta t$, the explicit Euler flow-matching update satisfies*

$$\max_k d_{\text{Euc}}\big(x_{t_k}^{\text{FM}}, x_{\text{geo}}(t_k)\big) \geq c \, \kappa_{\min} \, L^2 \Delta t \tag{11}$$

Lemma 2 highlights a complementary limitation to Lemma 1. Even in the absence of statistical noise, Euclidean straight-line parameterizations cannot faithfully track geodesic transport on curved manifolds: the deviation from the data manifold scales at least as $\Omega(\kappa_{\min}L^2)$ in continuous time, and as $\Omega(\kappa_{\min}L^2\Delta t)$ under few-step discretization. This formalizes the empirical observations from our spiral experiments in Section 4.1: Flow matching Lipman et al. (2022) and shortcut models Frans et al. (2024) systematically cut across the gaps between spiral arms and concentrate mass inside the spiral, rather than following the true

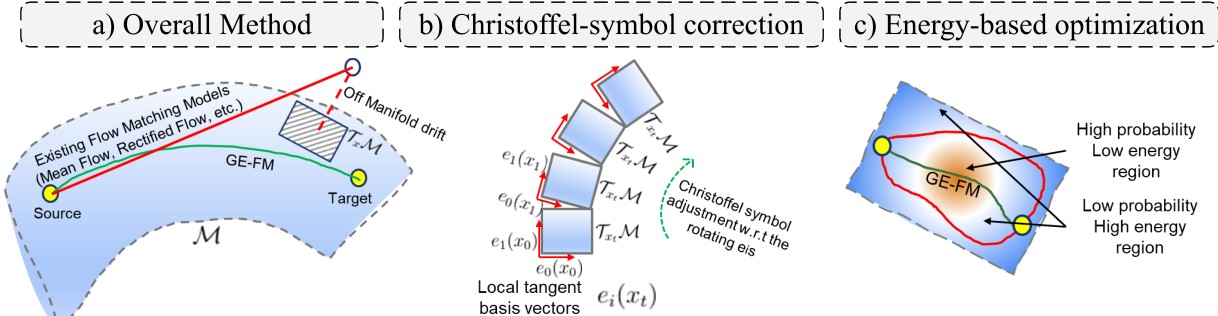

Figure 1: **Overview of GE-FM. (a) Limitation of Euclidean flow matching.** Standard methods use straight-line updates in ambient space, causing *off-manifold drift* and geometric mismatch when data lies on a curved manifold $\mathcal{M}$. GE-FM instead follows curvature-aware trajectories aligned with intrinsic geometry. **(b) Christoffel-symbol correction.** At each $x_t$, the velocity $v_\theta(x_t, t) \in T_{x_t}\mathcal{M}$. We induce a data-dependent metric $g(x, t)$ and compute Christoffel symbols $\Gamma(x, t)$, adding a correction term $\Gamma(x, t)[v_\theta, v_\theta]$ to account for curvature while remaining first-order. **(c) Energy-based stabilization.** Enforcing approximate conservation of intrinsic kinetic energy (under $g$) with a learned potential suppresses spurious acceleration, stabilizes large-step dynamics, and promotes low-energy transport paths.

curved manifold.[1] Combined with Lemma 1, this shows that neither purely first-order Euclidean objectives nor naive higher-order extensions provide a statistically and geometrically robust solution on highly curved data manifolds, motivating our energy-based, geometry-aware formulation.

## 3 Proposed Method

Keeping the limitations of existing flow matching techniques (Lemma 2, Lemma 1) in mind, we propose a **Geometry-aware Energy-optimized Flow Matching (GE-FM)** model that applies i) manifold curvature correction without explicitly modeling the higher-order terms (Section 3.1), ii) energy-based optimization to make the optimization landscape smoother and encourage lower energy paths to ensure better few-step performance (Section 3.2).

### 3.1 Geometry-aware dynamics via Christoffel correction

Lemma 1 shows that higher-order supervision is statistically much noisier than first-order velocity learning, especially when small temporal steps $h \ll 1$ are needed to resolve local dynamics. Lemma 2 then establishes that even with perfect velocities, Euclidean straight-line parameterizations incur an irreducible curvature bias of order $\Omega(\kappa_{\min}L^2)$ (or $\Omega(\kappa_{\min}L^2\Delta t)$ under few-step discretization) on curved manifolds. Together, these results imply that *neither* purely first-order Euclidean flows *nor* naive higher-order extensions provide a statistically and geometrically robust solution in the high-curvature regime. We now introduce a geometry-aware formulation that addresses the curvature bias of Lemma 2 *without* explicitly estimating higher-order derivatives, thereby sidestepping the variance blow-up from Lemma 1. The key idea is to endow the ambient space with a *learned Riemannian metric* induced by the velocity field, and to adjust the dynamics using the corresponding Christoffel symbols Do Carmo (2016); Lee (2003) so that transport follows geodesics of this metric. Existing approaches to geometry-aware transport largely operate through tangent space projections or diffeomorphic mappings from curved spaces to Euclidean coordinates (Section A.2). These methods assume access to a known manifold structure (either prescribed analytically or approximated locally) and correct the transport dynamics accordingly. However, in the few-step regime, local tangent space approximations accumulate curvature error over large intervals Lee et al. (2007), and diffeomorphic chart maps introduce distortions that compound when the step size spans a significant fraction of the trajectory. Moreover, these techniques are typically tied to specific geometric assumptions (hyperbolic, spherical, or product manifolds), limiting their applicability to settings where the underlying geometry of the transport path is neither known *a priori* nor fixed across the diffusion process Park et al. (2023); Roy et al. (2026b).

---

[1]Lemma 2 is stated for a smooth 1D embedded curve for clarity. The same argument extends to higher-dimensional submanifolds, replacing the scalar curvature $\kappa$ with the minimum principal curvature of the second fundamental form, giving the same structural bound $d_{\mathrm{Euc}}(x_{\mathrm{lin}}(t), \mathcal{M}) \geq c\,\kappa_{\min}\,t(1-t)\,L^2$. Our results on OASIS-3 ($d$=741, Section D) and MNIST ($d$=784, Section 4.2) confirm the mechanism operates well beyond the 1D setting.

**Induced metric from the velocity field.** Given a learned velocity field $v_\theta(x,t)$, we define a local, data-dependent metric

$$g(x,t) \;=\; I_d + \alpha\, J_v(x,t)^\top J_v(x,t) \tag{12}$$

where $J_v(x,t) = \nabla_x v_\theta(x,t)$ is the Jacobian of the velocity with respect to space and $\alpha > 0$ is a hyperparameter. Intuitively, $g$ amplifies directions in which the flow stretches or contracts the space, capturing the *effective* geometry induced by the transport. In regions where the dynamics bend strongly (e.g., along the outer arms of a spiral), the metric exhibits non-trivial curvature, while it remains close to Euclidean in nearly straight regions. $g$ can be interpreted as a Jacobian-induced pullback metric, where $J_v^\top J_v$ encodes local deformation of the learned transport, in line with prior work deriving data-dependent geometry from Jacobians in diffusion and representation spaces (Park et al., 2023; De Bortoli et al., 2022). Compared to existing manifold-aware flow matching approaches that rely on explicit charts or higher-order structure (Chen & Lipman, 2023; de Kruiff et al., 2024), this choice provides the lowest-complexity first-order, positive semi-definite approximation that is both computationally efficient and directly aligned with the learned dynamics.

**Christoffel symbols and intrinsic acceleration.** Here $g(x,t)$ is treated as a *family of spatial metrics* indexed by the diffusion/transport time $t$, not as a space-time metric on $\mathcal{M} \times [0,1]$. At each fixed $t = \tau$, we compute the Levi–Civita connection of the spatial metric $g(\cdot, \tau)$ and regularize the corresponding frozen-time spatial geodesic residual; we do not solve the exact space-time geodesic equation of a fully time-dependent action, which would introduce additional $\partial_t g$ terms. For a Riemannian metric $g$, the Christoffel symbols $\Gamma^k_{ij}(x,t)$ of the Levi–Civita connection are defined in local coordinates by

$$\Gamma^k_{ij} = \frac{1}{2} g^{k\ell}\big(\partial_i g_{\ell j} + \partial_j g_{\ell i} - \partial_\ell g_{ij}\big) \tag{13}$$

where $g_{ij}$ are the entries of the metric tensor and $g^{k\ell}$ are the entries of $g^{-1}$ and $\partial_i$ denotes partial differentiation with respect to $x_i$. Given a velocity $v \in \mathbb{R}^d$, and the Christoffel term $\Gamma(x,t)[v,v] \in \mathbb{R}^d$, then, $\big(\Gamma(x,t)[v,v]\big)^k = \sum_{i,j} \Gamma^k_{ij}(x,t)\, v^i v^j$, represents the intrinsic acceleration required for a curve with tangent $v$ to remain a geodesic under $g$.

**Geodesic residual and Christoffel-corrected dynamics.** On a Riemannian manifold $(\mathbb{R}^d, g)$, a geodesic $x^*(t)$ with tangent $v^*(t) = \frac{dx^*(t)}{dt}$ satisfies the geodesic equation

$$\frac{d^2 x^*(t)}{dt^2} + \Gamma\big(x^*(t), t\big)\big[v^*(t), v^*(t)\big] = 0 \tag{14}$$

Our learned dynamics evolve according to $\frac{dx(t)}{dt} = v_\theta(x(t),t)$, with material derivative $\frac{d^2 x(t)}{dt^2} = J_v(x(t),t)\, v_\theta(x(t),t)$. We therefore define the *geodesic residual*

$$R_\theta(x,t) \;=\; J_v(x,t)\, v_\theta(x,t) \;+\; \Gamma(x,t)\big[v_\theta(x,t), v_\theta(x,t)\big] \tag{15}$$

which measures the violation of the geodesic equation equation 14 under the induced metric $g$. Here $t$ is held fixed as a conditioning parameter when forming $\Gamma(x,t)$ and $J_v(x,t)$: equation 15 is a *frozen-time spatial* residual, not the material derivative of the non-autonomous sampling ODE $\frac{dx(t)}{dt} = v_\theta(x(t),t)$ with respect to $t$. In particular, no $\partial_t v_\theta$ term is omitted from the quantity being regularized, since $R_\theta$ does not differentiate with respect to the conditioning time $t$ itself. Our geometry-aware loss is

$$L_{\text{geo}}(\theta) \;=\; \mathbb{E}_{x,t}\big[\|R_\theta(x,t)\|_2^2\big] \tag{16}$$

added to the standard flow-matching objective. Minimizing $L_{\text{geo}}$ encourages the learned trajectories to be approximate geodesics with respect to $g$, aligning the dynamics with the curved transport geometry.

Crucially, both $g$ and $\Gamma$ in equation 12–equation 15 depend only on *first-order* spatial derivatives $J_v$. No explicit acceleration labels or higher-order temporal derivatives are required, so the estimator variance does not inherit the $h^{-2}$ dependence of Lemma 1. We now show that controlling the geodesic residual $R_\theta$ suffices to remove the dominant curvature bias identified in Lemma 2.

> **Lemma 3** (**Curvature bias reduction via Christoffel correction**). *Let the assumptions of Lemma 2 hold. Let $x^*(t)$ denote the (constant-speed) geodesic transport trajectory on $\mathcal{M}$, parametrized on $[0,1]$, with velocity $\frac{dx^*(t)}{dt}$. Let $x_\theta(t)$ be the trajectory induced by the learned velocity field $v_\theta(x,t)$, with induced metric $g$ and geodesic residual $R_\theta$ defined in equation 12–equation 15. Assume: (i) (**matching initial data**) $x_\theta(0) = x^*(0)$ and $\frac{dx_\theta(0)}{dt} = \frac{dx^*(0)}{dt}$; and (ii) (**boundedness / Lipschitzness in a tube**) both trajectories stay in a neighborhood $\mathcal{U}$ where the Christoffel quadratic form is Lipschitz: there exist constants $V, L_\Gamma < \infty$ such that for all $x,y \in \mathcal{U}$ and all $u,w \in \mathbb{R}^d$ with $\|u\|, \|w\| \le V$,*
>
> $$\left\| \Gamma(x,t)[u,u] - \Gamma(y,t)[w,w] \right\| \ \le \ L_\Gamma \big( \|x-y\| + \|u-w\| \big) \quad \forall t \in [0,1]. \tag{17}$$
>
> *Assume further that along the learned trajectory, $\|R_\theta(x_\theta(t),t)\|_2 \le \varepsilon \quad$ for all $t \in [0,1]$. Then there exists a constant $C > 0$ (depending only on $L_\Gamma$ and $V$, but not on $\varepsilon$) such that*
>
> $$\max_{t \in [0,1]} \|x_\theta(t) - x^*(t)\|_2 \ \le \ C\varepsilon \tag{18}$$

Lemma 3 (see Section G.4 for proof) shows that, when the geodesic residual is small, the learned Christoffel-corrected trajectories track true geodesics up to an $O(\varepsilon)$ error that does not grow with curvature or path length. This yields the following picture: (i) Naive higher-order models suffer from large statistical variance $\varepsilon_2 \gg \varepsilon_1$ (Lemma 1). (ii) Purely Euclidean first-order models suffer from curvature bias $\Omega(\kappa_{\min} L^2)$ in continuous time and $\Omega(\kappa_{\min} L^2 \Delta t)$ under few-step discretization (Lemma 2). (iii) GE-FM (Ours) uses only first-order information but enforces a small residual $R_\theta$, reducing curvature error to $O(\varepsilon)$ (Lemma 3), which matches the empirical improvements on curved spirals and irregular circles.

## 3.2 Energy-Based Stabilization

Section 3.1 resolved the dominant geometric failure mode of Euclidean flow matching. However, eliminating curvature bias alone does not ensure stable few-step transport. Even when trajectories are geometrically aligned with intrinsic geodesics, the learned velocity field $v_\theta(x,t)$ remains unconstrained in magnitude. In the few-step regime, where each ODE update spans a large interpolation length, uncontrolled velocity growth induces: i) energy drift along trajectories, ii) stiff dynamics under coarse discretization, iii) sensitivity to SGD noise and hyperparameter perturbations, and iv) sharp, anisotropic minima in parameter space Hairer et al. (2006); Battash et al. (2024); Tu (2022); An & Ying (2021); Chaudhari et al. (2019).

A naive remedy would be explicit higher-order supervision (acceleration and/or jerk). Yet Lemma 1 demonstrates that such supervision is statistically fragile, with variance scaling as $h^{-4}$ under finite-difference estimation. Thus, we require a mechanism that stabilizes dynamics without introducing noisy higher-order targets. We now introduce an energy-based principle that enforces this using first-order information. Rather than supervising acceleration explicitly, we regulate the dynamics through approximate conservation of total energy under the induced metric.

**Intrinsic kinetic energy under the induced geometry.** Recall from Section 3.1 that the Christoffel correction is defined with respect to the learned Riemannian metric Equation (12). This metric captures directions of local flow deformation and encodes the evolving geometry of the diffusion manifold. Under $g$, the intrinsic kinetic energy is $\mathrm{KE}_\theta(x,t) = \frac{1}{2}v_\theta(x,t)^\top g(x,t)v_\theta(x,t)$. Expanding, we get

$$\mathrm{KE}_\theta(x,t) = \frac{1}{2}\|v_\theta(x,t)\|^2 + \frac{\alpha}{2}\|J_v(x,t)v_\theta(x,t)\|^2 \tag{19}$$

The first term controls velocity magnitude. The second term penalizes variation of the velocity field along its own direction, which directly suppresses intrinsic acceleration $J_v v_\theta$, the same quantity appearing in the geodesic residual. Thus, kinetic energy regularization is not ad hoc; it couples directly to the curvature-aware structure introduced in Section 3.1.

**Total energy and conservative dynamics.** Controlling kinetic energy alone does not prevent cumulative drift. We therefore introduce a learned scalar potential $U_\theta(x,t)$ and define total energy $E_\theta(x,t) = \mathrm{KE}_\theta(x,t) + U_\theta(x,t)$.

In conservative mechanical systems, geodesics under a fixed metric preserve total energy. While exact conservation is neither expected nor required in discretized generative transport, approximate conservation imposes a powerful structural constraint: i) kinetic growth must be compensated by potential change, ii) intrinsic acceleration is indirectly bounded, iii) energy explosion is suppressed even under coarse $\Delta t$. This provides a first-order surrogate for higher-order stabilization without incurring the variance blow-up (Lemma 1).

**Energy conservation objective.** Let $x' = x + \Delta t \cdot v_\theta(x, t)$. We penalize finite-difference energy drift:

$$\mathcal{L}_E(\theta) = \mathbb{E}_{(x,t)} \left[ (E_\theta(x', t + \Delta t) - E_\theta(x, t))^2 \right] \implies E_\theta(x', t + \Delta t) = E_\theta(x, t) + \Delta t \frac{d}{dt} E_\theta(x, t) + \mathcal{O}(\Delta t^2) \quad (20)$$

so $\mathcal{L}_E$ approximates $\Delta t^2 \cdot \mathbb{E} \left[ \left( \frac{d}{dt} E_\theta(x, t) \right)^2 \right]$. Thus, $\mathcal{L}_E$ directly penalizes energy drift along trajectories. Since $\frac{d}{dt} E = v^\top g \frac{dv}{dt} + \frac{1}{2} v^\top \frac{dg}{dt} v + \frac{dU}{dt}$, bounding $\frac{dE}{dt}$ indirectly constrains intrinsic acceleration $\frac{dv}{dt}$. This ensures that the curvature-aware trajectories enforced by $\mathcal{L}_{\text{geo}}$ remain dynamically stable under large integration steps.

**Effect on optimization geometry.** Beyond dynamical stabilization, energy conservation reshapes the training landscape in parameter space as shown in Proposition 1 (see proof in Section G.5).

> **Proposition 1** (**Sufficient condition for flatter baselines under energy conservation**). *Let $\mathcal{L}_{\text{base}}(\theta) = \mathcal{L}_{\text{FM}}(\theta) + \lambda_{\text{geo}} \mathcal{L}_{\text{geo}}(\theta)$ and $\mathcal{L}_{\text{total}}(\theta) = \mathcal{L}_{\text{base}}(\theta) + \lambda_E \mathcal{L}_E(\theta)$. Assume $\mathcal{L}_{\text{base}}$ has a (possibly non-unique) set of local minimizers $\mathcal{S}$ and that $\mathcal{L}_{\text{base}}$ is $\mu$-strongly convex and twice continuously differentiable in a neighborhood of each $\theta \in \mathcal{S}$. Let $\hat{\theta}_E$ be a local minimizer of $\mathcal{L}_{\text{total}}$ for sufficiently small $\lambda_E$, and let $\hat{\theta}_0 \in \mathcal{S}$ denote a local minimizer attained when optimizing $\mathcal{L}_{\text{base}}$ alone. If in a neighborhood of $\mathcal{S}$ the spectral norm of the base Hessian is Lipschitz in $\theta$ and there exists $c > 0$ such that*
>
> $$\|\nabla_\theta^2 \mathcal{L}_{\text{base}}(\theta)\|_2 \le c_0 + c \, \mathcal{L}_E(\theta) \implies \|\nabla_\theta^2 \mathcal{L}_{\text{base}}(\hat{\theta}_E)\|_2 \le \|\nabla_\theta^2 \mathcal{L}_{\text{base}}(\hat{\theta}_0)\|_2 \quad (21)$$
>
> *then for sufficiently small $\lambda_E$.*

Importantly, this does *not* claim that flatter landscapes universally accelerate convergence. Rather, it formalizes that energy conservation imposes a structural constraint on trajectory evolution that suppresses rapid parameter-induced variations of the learned dynamics. In the few-step regime, such rapid variations translate directly into stiff ODE dynamics, anisotropic curvature in parameter space, and sensitivity to SGD noise. Reducing Hessian anisotropy yields: i) improved conditioning, ii) wider minima, iii) reduced sensitivity to hyperparameters, and iv) more stable few-step integration. This is precisely the behavior observed in Figure 5, where GE-FM exhibits smoother, more isotropic loss contours compared to FM.

**Unified perspective.** We can now reinterpret the full objective as -

$$\mathcal{L}(\theta) = \mathcal{L}_{\text{FM}} + \lambda_{\text{geo}} \mathcal{L}_{\text{geo}} + \lambda_E \mathcal{L}_E + \lambda_{KE} \text{KE}_\theta(x, t) \quad (22)$$

as addressing the two orthogonal failure modes identified in Section 2: i) **Geometric bias** (Lemma 2) → reduced by Christoffel correction (Lemma 3) ii) **Statistical fragility of higher-order supervision** (Lemma 1) → avoided entirely iii) **Dynamical stiffness under coarse discretization** → controlled via energy con-

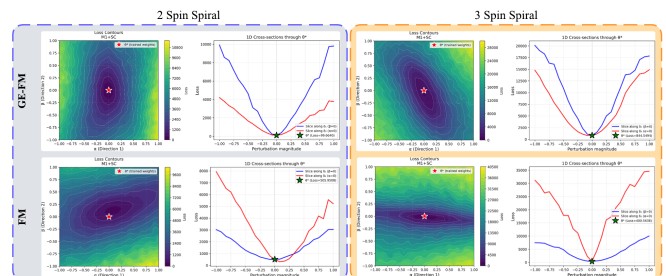

Figure 2: **Loss Landscape visualization.** GE-FM produces a smoother optimization landscape compared to FM.

servation (Proposition 1). Together, Christoffel correction enforces curvature-aware transport, while energy conservation ensures that such transport remains dynamically stable and well-conditioned in parameter space. The kinetic energy term penalizes variation in velocity magnitude along the trajectory, discouraging abrupt acceleration or deceleration that would cause large-step predictions to deviate from the true transport path. Crucially, all mechanisms rely only on first-order information, preserving statistical robustness while enabling stable few-step generation.

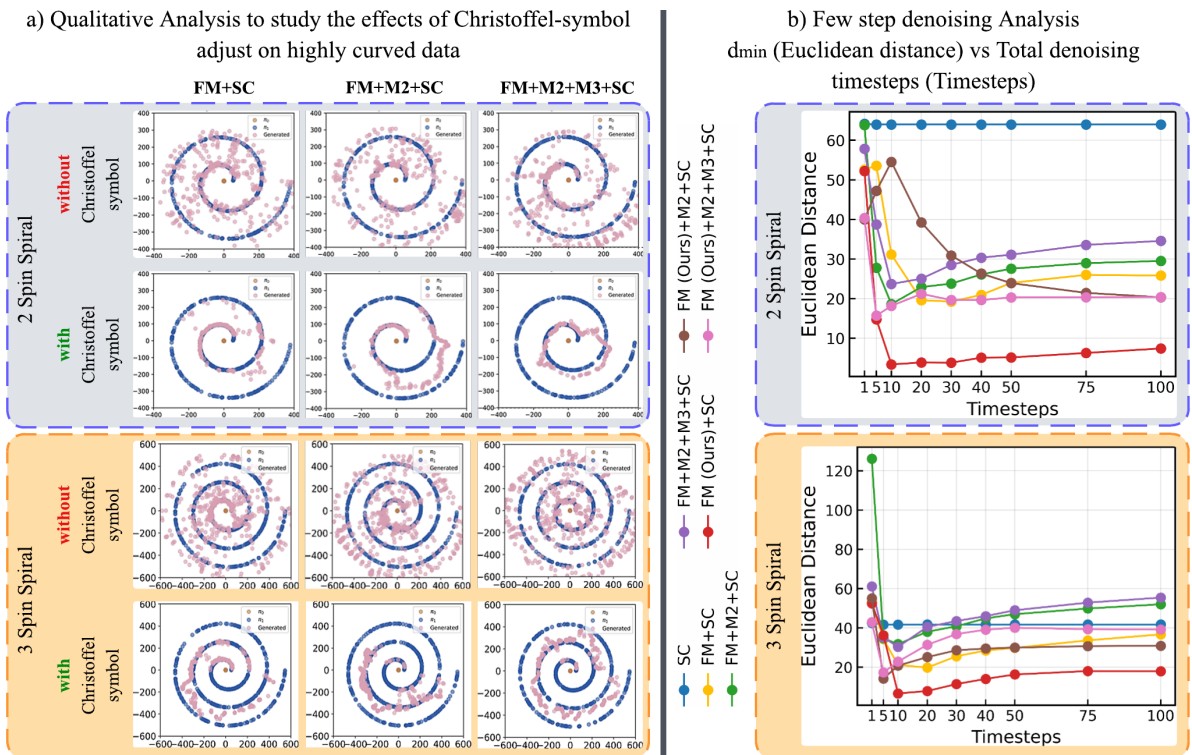

Figure 3: **Ablation studies on the impact of Christoffel-symbol correction.** a) Each panel shows samples from the source Gaussian (orange), the target spiral mixture (blue), and the terminal points of the learned transport (pink). b) Total denoising timestep (Timestep) vs $d_{min}$ (Euclidean distance) between the generated sample distribution (pink) and the target distribution (blue).

# 4 Experimental Results

We provide analysis on complex and highly curved synthetic spiral datasets (Section 4.1) showing the translation of theoretical validation to empirical results. Additionally, we empirically validate on three synthetic datasets: (i) irregular circle (Section C.3), ii) Gaussian 4-mode, and iii) 8-mode benchmarks (Section C.4). The evaluation protocol of the synthetic datasets is discussed in Section C.1. Lastly, we evaluate on the MNIST dataset to validate the generation capability in terms of Fretchet Inception Distance (FID) score. This extends the empirical validity of the proposed GE-FM and the superior theoretically grounded performance over the existing flow matching counterparts.

To validate that the geometric and energy-based mechanisms in GE-FM extend beyond synthetic curvature benchmarks to genuinely high-dimensional, real-world non-Euclidean data, we additionally evaluate on the OASIS-3 functional brain connectivity benchmark, where samples lie on the manifold of Symmetric Positive Definite matrices. Full experimental protocol, geometry-aware baseline comparisons, and quantitative results are reported in Appendix D. Synthetic dataset experiments were run on NVIDIA RTX A6000, whereas MNIST and OASIS experiments used NVIDIA H100.

## 4.1 Ablation on Complex Synthetic Datasets

The spiral dataset setting can be understood from Section C.2.

**Empirical analysis on synthetic datasets:** We study the effects of Christoffel-symbol correction and energy-based optimization in enhancing the performance of flow matching under highly curved settings. We further analyze the effects of energy-based batch reordering. Results demonstrate the superior few-step performance of the proposed GE-FM. We break down the analysis as follows:

*i) Few-step denoising and qualitative analysis comparing higher-order terms in flow matching vs Christoffel-symbol correction(Figure 3 a), b):* Higher-order terms (M2, M3) suffer due to the discretization error,

which piles up and degrades the performance even further. The proposed Christoffel-symbol correction achieves significantly better performance across all denoising timesteps. It also limits the performance degradation introduced by the higher-order terms. *Takeaway:* Purely first-order Euclidean objectives tend to underestimate curvature on these datasets and degrade sharply in the few-step regime (Lemma 2).

Higher-order modeling introduces further degradation (Lemma 1). Christoffel-symbol correction achieves geometry correction, which shows superior performance on highly curved benchmarks.

| Method | 3-Spin Spiral | | | | | 2-Spin Spiral | | | | |
|---|---|---|---|---|---|---|---|---|---|---|
| | 1 | 10 | 20 | 50 | 100 | 1 | 10 | 20 | 50 | 100 |
| Shortcut | **42.49** | 41.70 | 41.72 | 41.73 | 41.73 | 64.13 | 63.98 | 63.98 | 63.97 | 63.96 |
| Flow Matching | 51.55 | 49.41 | 34.62 | 27.19 | 29.82 | 53.36 | 36.35 | 24.03 | 17.86 | 19.47 |
| GE-FM (w/o batch reordering) | 49.75 | 14.53 | 6.05 | **4.75** | 7.09 | **51.41** | 7.72 | 2.58 | 2.41 | 2.40 |
| GE-FM (w batch reordering) | 56.82 | **2.46** | **2.23** | 2.46 | 3.42 | 50.11 | **3.81** | **1.60** | **1.95** | **2.38** |

Table 1: $d_{\min}$ ($\downarrow$) Equation (23) under varying ODE sampling steps on curved spiral benchmarks.

*ii) Loss landscape visualization to analyze the effect of energy-based optimization (Figure 2 a):* Following Li et al. (2018), we plot loss surfaces $\mathcal{L}(\theta^* + \alpha\delta_1 + \beta\delta_2)$ along random orthonormal directions around $\theta^*$. Energy conservation fundamentally reshapes the optimization geometry, yielding wider, flatter minima with improved conditioning. Standard FM converges to sharp, narrow valleys with tightly packed contours and steep walls. The landscape is highly anisotropic, with elongated elliptical contours indicating poor Hessian conditioning; cross-sections are asymmetric, with loss increasing by $\sim 8\times$ along one direction under unit perturbation while remaining flatter along the orthogonal direction—revealing a fragile, ridge-like solution sensitive to noise and hyperparameters. In contrast, GE-FM produces substantially flatter basins across both 2-spiral and 3-spiral tasks. The surface exhibits smooth slopes with widely spaced, near-circular contours, characteristic of isotropic and well-conditioned curvature. Cross-sections are symmetric and approximately parabolic, with gradients $\sim 3$–$5\times$ smaller than FM. This flattening persists across task complexity, indicating that energy conservation acts as a task-agnostic geometric regularizer. The resulting broad minima imply convergence to stable regions where perturbations (e.g., stochastic gradients, initialization, or hyperparameters) have limited impact. *Takeaway:* FM yields sharp, poorly conditioned minima with anisotropic curvature, whereas GE-FM enforces geodesic motion on a kinetic-energy manifold, acting as an implicit regularizer that smooths the loss landscape. This reduces effective degrees of freedom and produces minima that are $3$–$5\times$ flatter (in gradient magnitude and curvature), which is strongly associated with improved generalization and robustness to distribution shifts (Hochreiter & Schmidhuber, 1997; Keskar et al., 2016).

*iii) Qualitative and quantitative comparison with the existing flow matching methods (Figure 5, Section 4.1):* We visualize the generated output of FM (Flow matching), SC (Shortcut model), and GE-FM. We see significantly better alignment with the target distribution for GE-FM compared to FM and SC (Figure 5). Furthermore, we see in Section 4.1 that GE-FM outperforms FM and SC for all denoising timesteps. This demonstrates the effectiveness of the proposed method to address highly curved data manifolds. Furthermore, in Section 4.1 we see a significant boost in the performance with the application of batch-reordering (Section B): rather than using random

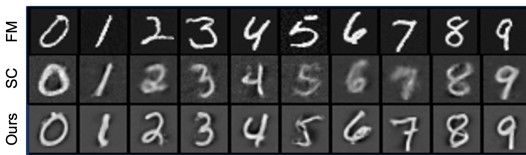

Figure 4: **Qualitative Analysis on MNIST dataset.** GE-FM achieves stronger sample quality in the single-step setting, attaining an FID of 130.91, while SC yields a higher (worse) FID of 158.80 under the same conditions. In contrast, standard FM is unable to produce meaningful images in a single step; using full 100-step inference, it reaches a lower FID of 118.95.

source-target pairs within a minibatch, we periodically (iterations 50-70% of training) re-pair sources and targets by solving a linear assignment problem that matches samples with similar total energy $E = \frac{1}{2}\|v\|_2^2 + U_\theta$, via the Hungarian algorithm. This discourages mismatched, "cross-spiral" pairings that would otherwise force the model to learn unrealistic velocity fields; full details and pseudocode are given in Algorithm 1 (Section B). *Takeaway:* Existing flow matching models underperform under non-Euclidean data manifolds, while the proposed GE-FM achieves superior performance even for few-step denoising. [2]

---

[2] Full coverage of highly curved, multi-turn manifolds in the one-step regime remains inherently difficult: large step sizes induce mode-averaging and trajectory collapse, a limitation shared by few-step flow-matching methods broadly rather than one specific to GE-FM. As shown in Section 4.1, coverage improves consistently as the number of denoising steps increases, indicating that Christoffel correction mitigates but does not entirely eliminate this fundamental expressivity limit of large-step integration.

## 4.2 Conditional generation with MNIST

MNIST provides a simple yet effective benchmark for conditional image generation, where each digit (0–9) defines a discrete conditioning variable corresponding to a distinct semantic class. This enables evaluation of both class consistency and visual coherence. We present one-step conditional samples (one per class) for each method. Standard flow matching (FM) fails to produce coherent outputs in a single step; thus, we use 100-step inference in Figure 4, where FM generates sharp, well-formed digits across all classes (FID = 118.95), serving as an upper bound despite minor artifacts. The short-cut model (SC) successfully generates all classes in one step, showing that self-consistency distillation transfers multi-step knowledge. However, samples are noticeably blurred, with loss of high-frequency details, especially for complex digits (5, 6, 7, 8), due to the mode-averaging effect of the $\ell_2$ objective, yielding washed-out outputs (FID = 158.80). In contrast, GE-FM produces one-step samples that are sharper

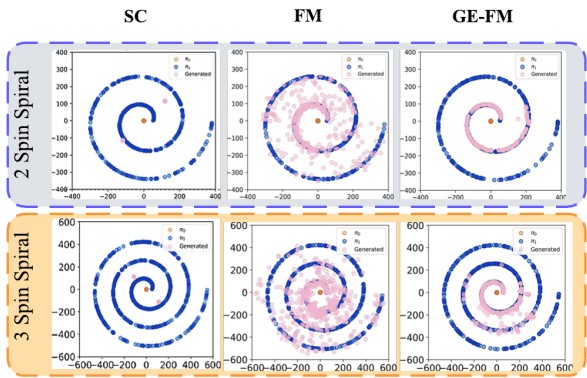

Figure 5: **Qualitative Analysis on Spiral datasets.** GE-FM produces visually far better samples matching the target distribution compared to FM and SC.

and more structurally coherent across all classes (FID = 130.91). Digits exhibit clearer boundaries, better contrast, and higher morphological diversity, with pronounced gains on complex digits (e.g., 5, 6, 8, 9). This improvement stems from the geometry-aware energy formulation, which enforces approximate energy conservation and induces curvature-adaptive (Christoffel-adjusted) dynamics. Consequently, GE-FM avoids $\ell_2$-induced smoothing, preserves high-frequency details, and achieves more faithful one-step generation.

## 5 Conclusion

This work revisits a fundamental question in flow-based generative modeling: *why do first-order Euclidean flows degrade under curvature and few-step discretization?* We uncover two coupled failure modes. First, Euclidean flow matching incurs an intrinsic curvature bias when transporting along straight-line interpolants, with deviation scaling as $\Omega(\kappa_{\min}L^2)$, which is exacerbated in the few-step regime due to coarse discretization. Second, explicit higher-order supervision is statistically fragile: finite-difference acceleration estimates suffer from variance blow-up scaling as $h^{-4}$, leading to instability.

To address this, we propose a unified first-order framework for curvature-aware, stable transport. (i) **Christoffel correction:** by inducing a data-dependent Riemannian metric and incorporating connection-level terms into the dynamics, we remove curvature bias without requiring tangent-space projections or global parameterizations, while remaining compatible with standard Euclidean networks. (ii) **Energy-based stabilization:** we introduce intrinsic kinetic energy under the induced metric and enforce approximate energy conservation, providing a first-order surrogate for higher-order control that suppresses acceleration and energy drift without statistical instability. This also biases optimization toward better-conditioned regions, improving robustness under coarse discretization.

Together, these components (see Section E for ablations) resolve the geometric and statistical limitations of Euclidean flow matching while preserving simplicity: the method remains first-order in time (no acceleration supervision), manifold-agnostic (no charts or projections), and effective in few-step regimes. Empirically, it yields consistent gains on both synthetic and real-world datasets, highlighting that geometry-aware corrections are most beneficial when curvature and discretization interact nonlinearly.

**Acknowledgements.** Prof. Lokhande acknowledges support from University at Buffalo startup funds, an Adobe Research Gift, an NVIDIA Academic Grant, and the National Center for Advancing Translational Sciences of the NIH (award UM1TR005296 to the University at Buffalo).

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

# Appendix

# A    Related Works

## A.1    Energy Matching

The connection between energy minimization and generative transport has deep roots in Optimal Transport theory. The Benamou-Brenier formulation Benamou & Brenier (2000) characterizes optimal transport as the minimization of total kinetic energy $\int_0^1 \int \|v_t(x)\|^2 \rho_t(x)\,dx\,dt$ subject to the continuity equation, establishing kinetic energy as the natural cost functional for measure transport. This variational perspective has directly influenced flow matching: OT-CFM Tong et al. (2023) and rectified flows Liu et al. (2022) encourage straight, low-energy trajectories that approximate the Benamou-Brenier solution. Action Matching Neklyudov et al. (2023) takes this further by learning a scalar potential $s(x, t)$ whose gradient defines a conservative velocity field, minimizing the Benamou–Brenier action functional from cross-sectional samples without requiring paired trajectories. In parallel, physics-informed architectures have explored energy conservation as a structural prior: Hamiltonian Neural Networks Greydanus et al. (2019) learn a scalar Hamiltonian whose symplectic gradient defines dynamics that exactly conserve energy, while Hamiltonian Generative Networks Toth et al. (2020) and Neural Hamiltonian Flows extend this to density modeling by using volume-preserving Hamiltonian dynamics as normalizing flows. Lagrangian Neural Networks Cranmer et al. (2020) complement these by learning Lagrangians directly, avoiding the requirement for canonical coordinates. Deep Lagrangian Networks Lutter et al. (2019) further demonstrate that Euler-Lagrange constraints can regularize learning to recover physically plausible dynamics, including the unsupervised decomposition of kinetic and potential energy.

More recently, energy-based perspectives have been applied directly to flow matching. Energy Matching Balcerak et al. (2025) unifies flow matching and energy-based models by learning a time-independent scalar potential that drives Optimal Transport far from the data manifold and transitions to a Boltzmann equilibrium near the data, achieving competitive generation quality on CIFAR-10 while retaining explicit likelihood information. Variational Potential Flow Bayes Loo et al. (2025) takes a related approach, learning an energy-parameterized potential flow matched to the data distribution through a KL-based variational loss without requiring Markov chain Monte Carlo (MCMC) sampling. The Kinetic Path Energy (KPE) diagnostic Li et al. (2026) analyzes flow matching through a kinetic energy lens, showing that per-sample accumulated kinetic energy correlates with both semantic fidelity and data density, and proposes Kinetic Trajectory Shaping (KTS) - a training-free inference procedure that modulates velocity magnitude across time to improve quality and reduce memorization. Physics-Constrained Flow Matching Utkarsh et al. (2025) enforces conservation laws and boundary conditions as hard constraints at inference time via projection onto constraint manifolds, though it operates post-hoc on pretrained models rather than during training. *Our work differs from all of the above in a key respect: rather than using energy as a generation mechanism (Energy Matching, VPFB), a diagnostic (KPE), or a post-hoc constraint (PCFM), we use energy conservation as a* training-time regularizer *for shortcut flow matching, directly targeting the quality degradation that arises in the few-step regime.*

## A.2    Manifold Aware Flow Matching

**Manifold-aware diffusion and score matching.** The geometry of latent space for diffusion models at each timestep of denoising is arbitrary, unlike other existing deep learning models Roy et al. (2026a); Liang et al. (2022); Liu et al. (2025). This adds to the challenge of manifold-aware diffusion denoising without having to explicitly guess or model the actual manifold. A large body of recent work extends score-based generative models beyond Euclidean data by formulating noising and denoising directly on Riemannian manifolds, which are suboptimal guesses of the unknown actual manifold structure. A representative example is Riemannian Score-Based Generative Modelling (RSGM) De Bortoli et al. (2022), which defines the forward process as a diffusion (e.g., Brownian motion) on a manifold and learns denoising scores compatible with the manifold geometry, including the use of intrinsic operators such as the Laplace-Beltrami operator and geometric connections encoded by the metric. Beyond fully intrinsic formulations, several practical works incorporate manifold-preserving corrections into diffusion sampling - often via projection, constraint, or

guidance steps designed to keep trajectories close to the data manifold Chung et al. (2022); He et al. (2023); Park et al. (2025).

**Manifold-aware flow matching.** In contrast to score-based models, there are limited works exploring manifold-aware flow matching. Riemannian Flow Matching (RFM) Chen & Lipman (2023) extends flow matching to general geometries by defining target vector fields through geometry-aware constructions and avoiding divergence computation on manifolds. Related work also investigates geometry-dependent interpolants and premetrics for flow matching objectives, including Metric Flow Matching, which designs smooth interpolations via metric structure Kapusniak et al. (2024). More recently, Pullback Flow Matching (PFM) emphasizes learning a latent geometry whose pullback preserves manifold structure, explicitly noting that requiring closed-form manifold mappings can be restrictive in practice de Kruiff et al. (2024). DiffeoCFM Collas et al. (2025) explored flow matching on non-Euclidean spaces by modeling a diffeomorphism from the curved manifold to $\mathbb{R}^2$

**Limitations of existing works in the few-step regime.** Despite promising progress, existing manifold-aware generative methods face practical and theoretical limitations when deployed in the *few-step* domain, where each update traverses a long interpolation length and local approximations can break down.

*(i) Locality of tangent-space / log–exp updates:* A common practice in manifold generative modeling is to alternate between tangent-space updates Roy et al. (2026b) and manifold projections using log and exp maps (or retractions). However, the exponential map is fundamentally a local construction: it is only well-behaved within a neighborhood (e.g., up to an injectivity radius), and can fail to be one-to-one beyond that region Jones (2024). Consequently, tangent-space approximations are most accurate for small step sizes and short transport lengths. In the few-step regime, where each update can traverse a large fraction of the trajectory, repeatedly relying on a locally valid tangent approximation can accumulate geometric distortion because the appropriate tangent space itself changes rapidly along curved paths.

*(ii) Dependence on global charts / diffeomorphic parameterizations:* Another line of work reduces manifold generation to Euclidean generation via an explicit mapping (or learned chart) between the manifold and a Euclidean latent space. While powerful when available, this approach requires a sufficiently global and well-conditioned parameterization. In general, not all manifolds admit a single global chart into $\mathbb{R}^m$, and even when a diffeomorphism exists on the relevant support, learning a mapping that is both stable and geometry-preserving (e.g., approximately isometric for transport) is a strong and task-dependent requirement. This limitation becomes pronounced for highly curved datasets and long interpolation lengths, where chart distortion can directly translate into sampling artifacts.

*(iii) Theoretical gaps for learned pullback geometry under coarse discretization:* Pullback-style approaches (e.g., PFM) learn a latent geometry whose pullback aims to respect manifold structure de Kruiff et al. (2024). While this provides a principled geometric viewpoint, existing formulations typically do not provide an explicit guarantee that few-step Euler integration of the learned first-order field remains close to an intrinsic geodesic when curvature is high. In particular, the dominant curvature deviation identified in Lemma 2 arises from missing connection-level correction terms; simply specifying a (pullback) metric does not by itself control the discretization error induced by coarse steps. This motivates incorporating curvature-aware corrections that directly target the acceleration term responsible for Euclidean shortcutting in few-step sampling.

*(iv) Metric choice and geometry mismatch in manifold diffusion guidance:* In manifold-aware diffusion and guidance-based corrections, the choice of geometry (metric, projection rule, or constraint manifold) is often problem-dependent: different metrics induce different notions of geodesics and different correction directions. While fully intrinsic diffusion (e.g., RSGM De Bortoli et al. (2022)) is well-defined once the manifold and metric are specified, in many practical pipelines the geometry used for correction is an approximation (e.g., VAE latent geometry or learned manifold surrogates), which may not align with the true transport geometry of the data. This can lead to guidance directions that preserve local validity but remain suboptimal under long-range transport. In contrast, our approach induces geometry directly from the learned dynamics and applies a connection-level correction that targets the specific curvature bias mechanism in Lemma 2. [3]

---

[3]While these methods focus on predefined or explicitly parameterized manifolds, our setting is fundamentally different: we consider manifold-agnostic, data-induced geometries that evolve with the learned dynamics, making direct empirical comparison with prior geometry-aware flow matching approaches neither well-posed nor informative.

**Advantages of the proposed Christoffel symbol adjustment.** Our method targets the precise mechanism identified in Lemma 2: curvature-induced deviation of Euclidean first-order transport under long interpolation lengths and coarse discretization. In contrast to tangent-space projection strategies, which rely on locally valid log–exp mappings, we operate directly in ambient space and introduce a connection-level correction via Christoffel terms induced by a learned, data-dependent metric. This explicitly compensates for the missing acceleration term in Euclidean flow matching, rather than approximating it through repeated local retractions.

Unlike pullback or diffeomorphic approaches that depend on learning a global chart or geometry-preserving mapping, our formulation does not require constructing an explicit manifold parameterization. Instead, the geometry is induced implicitly from the learned dynamics themselves through $g = I + \alpha J_v^\top J_v$, and the corresponding Christoffel symbols directly correct the second-order curvature term responsible for shortcutting effects. This avoids sensitivity to chart distortion and eliminates reliance on global coordinate constructions.

Furthermore, prior manifold-aware diffusion and flow methods typically modify the metric structure but do not explicitly control discretization-induced acceleration error in few-step generation. Our approach complements geometric correction with intrinsic kinetic-energy regularization under the same induced metric. This enforces stability of the learned geodesic dynamics and bounds stiffness without requiring explicit higher-order supervision (Lemma 1). The combination of connection-level correction and energy-based stabilization yields a transport mechanism that is both curvature-aware and dynamically stable. Recent geometric FM variants also differ fundamentally from our setting: some assume closed-form manifold operations and variational endpoint-based training (Zaghen et al., 2025), while others first infer a density/energy-defined metric and then optimize geodesic interpolants (Zweig et al., 2025). In contrast, we remain strictly first-order and manifold-agnostic, inducing geometry directly from the learned transport Jacobian.

Importantly, our framework is *manifold-agnostic* and *diffusion-agnostic*: it does not assume an intrinsic manifold representation, closed-form exponential maps, or a predefined Riemannian structure. Instead, geometry is learned implicitly from data and enforced directly at the level of the flow dynamics. This positions our method not merely as an extension of existing manifold diffusion or Riemannian flow techniques, but as a connection-aware reformulation of flow matching that directly addresses the curvature bias and few-step instability mechanisms formalized in Section 2.

**Comparison with Metric Flow Matching (MFM).** The structurally closest prior work is Metric Flow Matching (Kapusniak et al., 2024), which, like GE-FM, replaces straight-line Euclidean interpolation with a learned, data-dependent Riemannian metric; the two methods differ only in what that metric is learned from. MFM derives its metric from sample density, bending interpolants toward where data is concentrated, whereas GE-FM induces the metric directly from the velocity field's own Jacobian ($g = I + \alpha J_v^\top J_v$) and corrects the dynamics via a Christoffel connection term that follows the curvature the flow itself implies, eliminating MFM's upfront density-estimation stage. On the OASIS-3 functional connectivity benchmark (Section D), GE-FM improves on MFM across every metric with consistently tighter variance, since MFM's density-driven metric discourages interpolants from reaching the sparse tails of the distribution and thus caps coverage (Table 3).

# B  Energy-Based Batch Reordering

Standard flow matching relies on random source-target pairings within each minibatch, effectively sampling from the independent coupling $p_0 \times p_1$. While unbiased in expectation, such random pairing can induce highly unnatural transport directions in practice, particularly on curved data manifolds where Euclidean proximity does not imply geodesic closeness. In the few-step regime, these mismatched pairings exacerbate curvature-induced deviation, leading to unstable or inefficient vector field learning Pooladian et al. (2023).

Recent works attempt to improve pairing through optimal transport (OT)-based couplings, either via minibatch OT Tong et al. (2023) or via globally precomputed transport maps Kornilov et al. (2024). However, minibatch OT provides only a noisy local approximation of the global geometry and is sensitive to batch size Nguyen et al. (2022), while global OT-based alignment fixes the coupling throughout training and may not adapt to evolving learned dynamics Genevay et al. (2019). Furthermore, global alignment strategies such

as AlignFlow Kong et al. (2025) construct a deterministic noise–data coupling using semi-discrete optimal transport (SDOT) computed once before training. Such approaches improve trajectory straightness by reducing Euclidean transport cost but do not account for the evolving learned dynamics or energy landscape of the model.

To address this gap, we introduce an *energy-based batch reordering* procedure that dynamically realigns source–target pairs during training. Rather than minimizing Euclidean transport cost, we minimize energy mismatch under the learned dynamics. This encourages physically consistent transport paths aligned with the learned energy landscape, which is especially important under curvature and few-step discretization.

---

**Algorithm 1** Energy-Based Batch Reordering

---

**Require:** Source batch $\{z_0^i\}_{i=1}^B$, target batch $\{z_1^j\}_{j=1}^B$, model $\theta$, time $d$

**Ensure:** Reordered targets $\{z_1^{\pi(i)}\}_{i=1}^B$

1: $v_0, \_\_ \leftarrow \theta(z_0^i, 0, d)$     ▷ Velocity at $t = 0$
2: $v_1, \_\_ \leftarrow \theta(z_1^j, 1, d)$     ▷ Velocity at $t = 1$
3: $E_0^i \leftarrow \frac{1}{2}\|v_0^i\|_2^2 + U_\theta(z_0^i, 0, d)$     ▷ Source energy
4: $E_1^j \leftarrow \frac{1}{2}\|v_1^j\|_2^2 + U_\theta(z_1^j, 1, d)$     ▷ Target energy
5: $C_{ij} \leftarrow (E_0^i - E_1^j)^2$     ▷ Cost matrix
6: $\pi \leftarrow$ linear_sum_assignment($C$)     ▷ Optimal matching
7: **return** $\{z_1^{\pi(i)}\}_{i=1}^B$

*Key insight:* Energy conservation $E(t) \approx$ const implies that optimal source–target pairs should have matched total energies $E_0 \approx E_1$. By solving the linear assignment problem

$$\pi^* = \arg\min_\pi \sum_i (E_0^i - E_1^{\pi(i)})^2,$$

We obtain pairings that respect the learned energy landscape while avoiding long-range Euclidean shortcuts.

---

Algorithm 1 is applied mid-training (iterations 50-70%) to a random minibatch of source-target pairs $\{z_0^i, z_1^j\}$: (i) predict velocities $v_\theta(z, t)$ at $t = 0, 1$; (ii) compute total energies $E = \frac{1}{2}\|v\|_2^2 + U_\theta(z, t)$ using the learned potential $U_\theta$; (iii) solve the assignment problem via the Hungarian algorithm to obtain reordered targets $z_1^{\pi(i)}$. This procedure (i) enforces energy consistency along transport paths (ii) reduces curvature-induced artifacts from mismatched pairings, and (iii) incurs negligible overhead ($O(B^2)$ solvable in $O(B^3)$) Section 4.1 shows that energy-based reordering provides a substantial boost to few-step performance on curved benchmarks, complementing our Christoffel correction by ensuring geometrically meaningful training pairs from the outset.

## C Synthetic dataset experiments

### C.1 Evaluation Protocol

To evaluate transport quality, we use the minimum Euclidean distance from each generated sample $z_T \sim \pi_1^{\text{flow}}$ to the set of true spiral vertices $\{v_k\}_{k=1}^K$, where each $v_k = (r(\theta_k)\cos\theta_k, \, r(\theta_k)\sin\theta_k)$ denotes the mean of the $k$-th Gaussian component in $\pi_1$. Specifically, we report the batch-average as shown in Equation (23), where $B$ is the number of generated samples and $\|\cdot\|_2$ is the Euclidean norm. Lower $d_{\min}$ indicates better mode coverage and alignment with the true spiral manifold.

$$d_{\min}(z_T) = \frac{1}{B} \sum_{i=1}^B \min_{1 \leq k \leq K} \|z_T^{(i)} - v_k\|_2 \tag{23}$$

### C.2 Spiral Datasets

The spiral benchmarks are constructed by taking a simple 2D Gaussian source distribution $\pi_0$, centered at the origin with isotropic covariance, and defining a target $\pi_1$ as a Gaussian mixture whose component means lie along a multi-turn spiral in the plane. A radius function $r(\theta)$ (e.g., with slightly super-linear growth in $\theta$) is specified, a set of angles $\{\theta_k\}$ is discretized over several full rotations, and each spiral point $(r(\theta_k)\cos\theta_k, \, r(\theta_k)\sin\theta_k)$ is used as the mean of an isotropic Gaussian component. This yields a highly curved, non-Gaussian target where density winds around the origin multiple times with increasing radius.

A key difficulty is that the *intrinsic* along-manifold distance between two spiral points can be large even when their Euclidean distance $d_{\text{Euc}}(x,y) = \|x - y\|_2$ is small, because different turns of the spiral pass close to each other in $\mathbb{R}^2$. As a result, straight-line transport from $\pi_0$ to the outer arms would have to cut through extensive low-density regions, and methods that implicitly assume near-linear trajectories in the ambient Euclidean metric tend to either under-reach the outer modes or

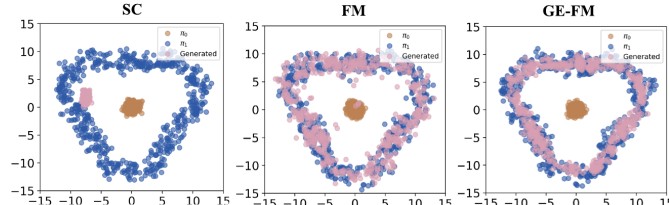

Figure 6: **Qualitative Analysis on Irregular Circle datasets.** GE-FM produces visually far better samples matching the target distribution compared to FM and SC.

collapse mass toward the center instead of following the true curved manifold. In this work, we use these spiral datasets to probe how Christoffel-symbol corrections and energy-based optimization influence the learned transport geometry and robustness of rectified-flow models in such strongly non-Euclidean settings.

### C.3 Irregular Circle Datasets

**Irregular Circle Dataset:** The irregular circle benchmark is constructed to test transport methods on targets with continuous non-uniform curvature and local geometric irregularities. The source distribution $\pi_0$ is a simple 2D isotropic Gaussian centered at the origin with covariance $\sigma^2 I$, representing a concentrated mass. The target distribution $\pi_1$ is defined as a Gaussian mixture whose component means lie along an *irregular circle* in the plane, created by perturbing a base circular trajectory with deterministic and stochastic variations in radial distance. Specifically, we discretize $N$ angles $\theta_k = 2\pi k/N$ for $k = 0, \ldots, N-1$ around the origin, and for each angle compute a perturbed radius $r(\theta_k) = D + A\sin(m\theta_k) + \epsilon_k$, where $D$ is the base radius, $A\sin(m\theta_k)$ introduces a deterministic $m$-fold sinusoidal modulation (e.g., $m = 3$ for a trefoil-like pattern), and $\epsilon_k \sim \text{Uniform}(-\delta, \delta)$ adds random local irregularities. Each mixture component is then an isotropic Gaussian with mean $\big(r(\theta_k)\cos\theta_k,\ r(\theta_k)\sin\theta_k\big)$ and shared variance $\sigma^2$. This construction yields a target that is neither perfectly circular nor chaotic, but rather exhibits *structured irregularity*: the sinusoidal modulation creates predictable large-scale variations in curvature, while the random perturbations introduce small-scale roughness that breaks any exact symmetry.

The resulting transport problem is challenging because straight-line paths from the Gaussian center must navigate to a boundary whose local curvature and density vary continuously, and methods that rely on global geometric assumptions (e.g., convexity, uniform curvature) or that fail to capture fine-grained geometric structure may either under-reach certain regions of the irregular circle or exhibit non-uniform coverage. We use this irregular circle dataset to evaluate how higher-order rectified flows, self-consistency regularization, and geometric corrections handle continuous non-uniform curvature and stochastic geometric perturbations in optimal transport problems.

| Method | 1 | 10 | 20 | 50 | 100 |
|---|---|---|---|---|---|
| Shortcut | **1.09** | 1.80 | 1.82 | 1.84 | 1.84 |
| Flow Matching | 6.37 | 0.82 | 0.66 | 0.57 | 0.55 |
| GE-FM | 6.35 | **0.48** | **0.43** | **0.42** | **0.42** |

Table 2: $d_{\min}$ ($\downarrow$) Equation (23) under varying ODE sampling steps on the Irregular Circle benchmark.

**Empirical Results:** As seen in Section C.3, GE-FM outperforms FM and SC significantly as the denoising timesteps increase. Furthermore, qualitative results (Figure 6) illustrate better alignment with the target distribution. This strengthens the claimed advantages of GE-FM over the existing flow matching counterparts.

### C.4 Gaussian Synthetic Datasets

**Multi-Mode Gaussian Datasets:** The multi-mode Gaussian benchmarks are constructed by placing discrete Gaussian mixture components at the vertices of regular polygons in 2D, creating challenging multi-modal transport problems with explicit rotational structure. For the source distribution $\pi_0$, we position $K$ isotropic Gaussian components (with variance $\sigma^2$) at radius $D_0$ from the origin, equally spaced at angles $\theta_k = 2\pi k/K$ for $k = 0, \ldots, K-1$, forming a symmetric arrangement (e.g., a square for $K = 4$ or an octagon for $K = 8$). The target distribution $\pi_1$ is similarly constructed but with components placed at a larger radius $D_1 > D_0$ and rotated by an offset angle $\phi$ (e.g., $\phi = \pi/4$ for the 4-mode case), yielding $\pi_1$

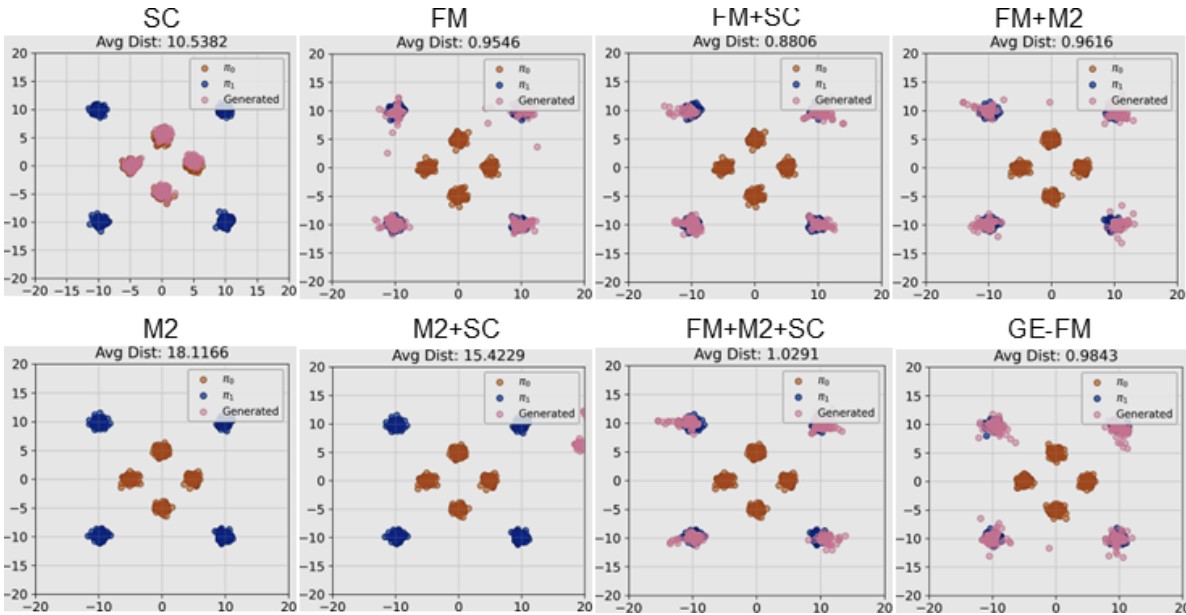

Figure 7: **Qualitative Analysis on Gaussian datasets.** Each panel shows samples from the source (orange), the target (blue), and the terminal points of the learned transport (pink).

components at positions $\big(D_1 \cos(\theta_k + \phi),\ D_1 \sin(\theta_k + \phi)\big)$. This design creates a fundamental ambiguity for transport methods: while the Euclidean distance between any source mode and its nearest target mode may be relatively small, the *optimal assignment* of source modes to target modes is non-trivial due to the rotation and scaling. Naive point-to-point matching can lead to trajectories that cross each other or traverse large angular displacements, whereas the true optimal transport should respect the underlying rotational symmetry. Furthermore, straight-line interpolation in ambient space must navigate the substantially low-density regions between modes, making this benchmark sensitive to whether a method can learn curved, mode-respecting trajectories versus collapsing toward straight paths that cut through probability voids. We use these multi-mode Gaussian datasets to evaluate how higher-order rectified flows, self-consistency regularization, and geometric corrections handle discrete multi-modal structure and rotational coupling in transport problems.

**Empirical Results:** GE-FM achieves performance that is comparable to Flow Matching (FM) and Shortcut (SC), rather than exhibiting the large improvements observed on the spiral and irregular-circle benchmarks. This behavior is consistent with the theoretical picture developed in Section 2. The transport geometry is approximately Euclidean. Each mode is locally isotropic, and the displacement between corresponding source and target modes is close to linear in ambient space. As a result, the curvature term $\kappa_{\min}$ appearing in Lemma 2 is effectively small in the regions traversed by optimal transport trajectories. Consequently, the curvature-induced bias term $\Omega(\kappa_{\min} L^2)$ becomes negligible, and straight-line Euclidean parameterizations already provide a good approximation to the intrinsic transport path. Since the dominant error mechanism identified in Lemma 2 is weak in this regime, the geometric correction provided by Christoffel adjustment does not yield dramatic improvements, as there is little curvature bias to remove. Importantly, however, GE-FM remains fully competitive with FM and SC across all sampling steps. This demonstrates that the geometry-aware and energy-based regularization does not introduce unnecessary bias or degradation when the data manifold is close to Euclidean. This robustness contrasts sharply with the highly curved spiral and irregular-circle benchmarks (Section 4.1 and **??**), where $\kappa_{\min}$ is strictly positive and often large. In those regimes, the curvature bias predicted by Lemma 2 becomes significant, and Euclidean methods exhibit substantial performance degradation, particularly in the few-step setting. GE-FM, by contrast, maintains stable and accurate transport, consistent with the $O(\varepsilon)$ tracking guarantee of Lemma 3. *Takeaway:* Results confirm that GE-FM adapts to the intrinsic geometry of the transport problem: it provides substantial gains in high-curvature regimes while remaining comparable to standard flow matching when curvature effects are minimal.

# D   Evaluation on the OASIS-3 Functional Brain Connectivity Benchmark

To demonstrate that GE-FM extends beyond synthetic curvature benchmarks to high-dimensional real-world non-Euclidean data, we evaluate on resting-state fMRI data from the OASIS-3 cohort (LaMontagne et al., 2019). The task is class-conditional generation of functional connectivity matrices: Symmetric Positive Definite (SPD) covariance matrices estimated from fMRI time series over 39 brain regions, conditioned on diagnosis (healthy vs. Alzheimer's). This setting combines strong non-linearity (SPD manifold structure) with the data scarcity typical of medical imaging.

## D.1   Experimental Protocol

We follow the evaluation protocol of DiffeoCFM (Collas et al., 2025) to ensure a fair comparison: MSDL atlas parcellation, 200 training epochs, normalized Cholesky map diffeomorphism embedding, and 10-fold GroupShuffleSplit cross-validation with subject-disjoint splits. Each SPD matrix is embedded into Euclidean space via the normalized Cholesky map diffeomorphism (a bijection to the strictly lower-triangular Cholesky factor), standardized, and passed to the generative model. For generation, outputs are mapped back to the SPD manifold via the inverse diffeomorphism.

**Metrics.**   We report $\alpha$-Precision and $\beta$-Recall (fidelity and diversity of the generated distribution), their harmonic mean $\alpha, \beta$-F1, and Classifier Adversarial Score (CAS) metrics (ROC-AUC and F1), which measure whether a classifier trained solely on synthetic data generalizes to held-out real data, directly probing downstream utility. All metrics are averaged over the 10 splits (mean ± std).

**Baselines.**   We compare against: (i) *Real Data* (oracle), where a classifier is trained and tested on real data; (ii) *TriangCFM*, a geometry-agnostic flow matching baseline in the lower-triangular embedding; (iii) *DiffeoGauss*, a class-conditional Gaussian in the normalized Cholesky map space capturing first- and second-order statistics; (iv) *DiffeoCFM* (Collas et al., 2025), manifold-aware conditional flow matching; and (v) $\text{MFM}_{\text{RBF}}$, Metric Flow Matching (Kapusniak et al., 2024), which also learns a data-dependent Riemannian metric for flow matching and is the most structurally similar prior work to GE-FM.

**GE-FM configuration.**   GE-FM requires no diffeomorphic preprocessing inherently; however, for fair comparison with DiffeoCFM we apply the same normalized Cholesky map embedding. GE-FM augments standard CFM with a *manifold-aware yet manifold-agnostic* penalty induced by the learned metric $g$, and corrects sampling via the Christoffel term $-\Gamma[v, v]$, encouraging geodesic-consistent transport. To address the $\mathcal{O}(d^4)$ cost of the exact Christoffel tensor, we use the contraction identity $\Gamma[v, v] = \frac{1}{2}g^{-1}(2A - C)$ described in Appendix F, which reduces computation to $2d + 1$ backward passes and memory from $\mathcal{O}(Bd^4)$ to $\mathcal{O}(Bd^2)$ without approximation. This is critical for OASIS-3, where the normalized Cholesky map embedding produces $d = 39 \rightarrow 741$ features and the naive tensor form causes OOM even on a 49 GB A6000 GPU.

## D.2   Quantitative Results

| Method | $\alpha$-Precision ↑ | $\beta$-Recall ↑ | $\alpha, \beta$-F1 ↑ | ROC-AUC ↑ | F1 ↑ |
|---|---|---|---|---|---|
| Real Data | $0.89 \pm 0.04$ | $0.88 \pm 0.05$ | $0.88 \pm 0.01$ | $0.71 \pm 0.04$ | $0.61 \pm 0.06$ |
| TriangCFM | $0.22 \pm 0.27$ | $0.26 \pm 0.37$ | $0.23 \pm 0.32$ | $0.51 \pm 0.05$ | $0.45 \pm 0.17$ |
| DiffeoGauss | $0.52 \pm 0.06$ | $0.31 \pm 0.02$ | $0.39 \pm 0.03$ | $0.68 \pm 0.03$ | $0.40 \pm 0.09$ |
| DiffeoCFM | $\mathbf{0.63 \pm 0.06}$ | $0.38 \pm 0.04$ | $0.47 \pm 0.05$ | $0.68 \pm 0.03$ | $0.53 \pm 0.04$ |
| $\text{MFM}_{\text{RBF}}$ | $0.52 \pm 0.01$ | $0.36 \pm 0.03$ | $0.45 \pm 0.02$ | $0.63 \pm 0.07$ | $0.50 \pm 0.05$ |
| **GE-FM (ours)** | $0.55 \pm 0.01$ | $\mathbf{0.42 \pm 0.01}$ | $\mathbf{0.48 \pm 0.01}$ | $\mathbf{0.71 \pm 0.04}$ | $\mathbf{0.55 \pm 0.07}$ |

Table 3: Quantitative comparison on OASIS-3. Mean ± std over 10 subject-disjoint splits. Best non-oracle results in **bold**.

**Comparison against DiffeoCFM.**   On OASIS-3, GE-FM improves $\alpha, \beta$-F1 over DiffeoCFM with approximately 5× lower variance, indicating substantially more stable generation across subject splits. GE-FM also

achieves the highest ROC-AUC, matching the Real Data oracle, while simultaneously improving CAS-F1. The trade-off is lower $\alpha$-Precision, as Christoffel correction spreads generated mass more uniformly along the manifold, boosting $\beta$-Recall at the cost of slight deviations from high-density regions. Thus, DiffeoCFM is sharper but mode-seeking, whereas GE-FM provides broader, more representative coverage; the improved CAS metrics confirm that the added diversity is meaningful rather than noise. TriangCFM collapses on this benchmark, underscoring the importance of geometry-aware modeling for SPD-valued data. Although DiffeoGauss achieves competitive ROC-AUC, its low recall highlights the limitations of unimodal Gaussian assumptions for modeling multimodal neuroimaging data.

**Comparison against Metric Flow Matching (MFM).** Both GE-FM and MFM replace straight-line Euclidean interpolation with a learned, data-dependent Riemannian metric, differing only in what that metric is learned from. MFM derives it from sample density, bending interpolants toward where the data is concentrated. GE-FM, by contrast, induces the metric directly from the velocity field's own Jacobian (via $g = I + \alpha J_v^\top J_v$) and corrects the transport dynamics via a Christoffel connection term, following the curvature that the flow itself implies. In Table 3, GE-FM improves on MFM across every metric and does so with consistently tighter variance ($\pm 0.01$ on three of five metrics).

This gap has a structural explanation. MFM's density-driven metric implicitly discourages interpolants from reaching the sparse tails of the distribution, capping its coverage (the $\beta$-Recall gap), which in turn yields a synthetic training set that misses parts of the real class manifold and transfers worse downstream (the 0.08 ROC-AUC gap, with GE-FM matching real data exactly). GE-FM's dynamics-induced geometry carries no such density bias and reads transport curvature directly, while also eliminating MFM's noisy upfront metric-estimation stage.

**Takeaway.** These results validate that GE-FM's curvature-aware correction provides meaningful gains in genuinely high-dimensional, real-world non-Euclidean settings ($d = 741$ after embedding) - not just on the low-dimensional synthetic curvature benchmarks in Section 4.1. The combination of broader coverage, tighter variance, and best-in-class downstream classification performance (matching the real-data oracle) demonstrates that the geometric mechanism scales beyond the toy regime.

## E   Ablation Experiments

**Analysis of computational overhead.** We report runtimes for the 1-step setting on two curved benchmarks in Table 4. GE-FM induces only additional training cost, while inference remains close to baselines ($<1$s). The geometry-aware and energy-based components introduce extra computations during optimization, whereas inference still uses a first-order transport model. Moreover, batch reordering further increases training time due to explicit matching/reassignment, but offers a small speedup at inference. This is a direct consequence of improved

| Dataset | Method | Ver. | Train | Test |
|---------|--------|------|-------|------|
| Irr. Circle | FM | – | 28.95 | 0.65 |
| | SC | – | 31.02 | 0.71 |
| | GE-FM (Energy optimization) | w/o batch reordering | 421.46 | 0.88 |
| | GE-FM (Energy optimization) | w batch reordering | 1382.85 | 0.80 |
| | GE-FM (Christoffel correction) | w/o batch reordering | 530.89 | 0.67 |
| | GE-FM (Christoffel correction) | w batch reordering | 1549.45 | 0.79 |
| | GE-FM | w/o batch reordering | 525.18 | 0.87 |
| | GE-FM | w batch reordering | 1604.65 | 0.82 |
| 3-spin Spiral | FM | – | 28.37 | 0.67 |
| | SC | – | 26.56 | 0.65 |
| | GE-FM (Energy optimization) | w/o batch reordering | 406.15 | 0.88 |
| | GE-FM (Energy optimization) | w batch reordering | 1112.47 | 0.83 |
| | GE-FM (Christoffel correction) | w/o batch reordering | 551.48 | 0.94 |
| | GE-FM (Christoffel correction) | w batch reordering | 1275.62 | 0.81 |
| | GE-FM | w/o batch reordering | 503.27 | 0.91 |
| | GE-FM | w batch reordering | 1372.39 | 0.84 |

Table 4: Training and test times (seconds) for 1-step setting.

source–target alignment during training, helping the model converge to a smoother transport map. *Trade-off*: increased training overhead in exchange for improved geometric fidelity and transport quality, with negligible deployment-time cost.

**Analysis of individual components.** Table 5 isolates the contributions of GE-FM's two components.

Across both datasets, the full model achieves the lowest $d_{\min}$, indicating that energy-based stabilization and Christoffel correction are complementary. The former improves large-step stability, while the latter enhances geometric alignment with curved trajectories. Our results on the challenging 3-spiral benchmark, which is highly curved, validate our claim. Batch reordering (Table 1, main paper) further improves performance with increasing denoising steps, owing to better source–target alignment.

| Dataset | Method | $d_{\min}$ |
|---|---|---|
| Irr. Circle | GE-FM (Energy optimization) | 6.49 |
| | GE-FM (Christoffel correction) | 6.42 |
| | GE-FM | 6.35 |
| 3-Spin Spiral | GE-FM (Energy optimization) | 51.75 |
| | GE-FM (Christoffel correction) | 53.10 |
| | GE-FM | 49.75 |

Table 5: $d_{\min}$ for 1-step setting comapring the components of GE-FM.

## F  Algorithmic overview of GE-FM and Additional Explanation of Design Choices

We represent the entire training and sampling loop in Algorithm 2. We also summarize the design choices behind the contraction identity used to compute the Christoffel correction efficiently and the role of the learned potential $U_\theta$ as an energy buffer.

---

**Algorithm 2** GE-FM: Unified Training and Sampling Procedure

> **Training**
> **Require:** Data distributions $\pi_0, \pi_1$; velocity network $v_\theta$; potential network $U_\theta$; metric hyperparameter $\alpha$; loss weights $\lambda_{\text{geo}}, \lambda_E, \lambda_{KE}$ (set via grid search); batch reordering window (iterations 50–70% of training)
>  1: **for** each training iteration **do**
>  2:   Sample minibatch $\{x_0^{(i)}\} \sim \pi_0$, $\{x_1^{(i)}\} \sim \pi_1$, $t^{(i)} \sim \mathcal{U}[0,1]$
>  3:   **if** current iteration $\in$ reordering window **then**
>  4:     Reorder targets via Algorithm 1
>  5:   **end if**
>  6:   Form interpolant $x(t) = \psi(x_0, x_1, t)$ and ground-truth velocity $v^*(x(t), t)$
>  7:   Compute $v_\theta(x(t), t)$ and base flow-matching loss $L_{\text{FM}}$ (Eq. 1)
>  8:   Compute Jacobian $J_v(x(t), t) = \nabla_x v_\theta(x(t), t)$
>  9:   Induce metric $g(x, t) = I_d + \alpha\, J_v^\top J_v$ (Eq. 12)
> 10:   Compute Christoffel contraction $\Gamma(x,t)[v,v] = \frac{1}{2}\, g^{-1}(2A - C)$ via $2d+1$ backward passes through $v_\theta$
> 11:   Compute geodesic residual $R_\theta$ (Eq. 15) and geometric loss $L_{\text{geo}}$ (Eq. 16)
> 12:   Compute intrinsic kinetic energy $KE_\theta(x,t)$ (Eq. 19) and total energy $E_\theta = KE_\theta + U_\theta$
> 13:   Compute energy-drift loss $L_E$ (Eq. 20)
> 14:   Form total objective $L(\theta) = L_{\text{FM}} + \lambda_{\text{geo}} L_{\text{geo}} + \lambda_E L_E + \lambda_{KE} KE_\theta(x,t)$ (Eq. 22)
> 15:   Update $\theta$ via gradient descent on $L(\theta)$
> 16: **end for**
> **Sampling**
> **Require:** Trained $v_\theta$; number of steps $N$
> 17: Initialize $x_0 \sim \pi_0$, $\Delta t = 1/N$
> 18: **for** $k = 0, \ldots, N-1$ **do**
> 19:   $x_{t_{k+1}} \leftarrow x_{t_k} + \Delta t \cdot v_\theta(x_{t_k}, t_k)$ ▷ standard first-order Euler integration; identical in implementation to baseline flow matching, no additional geometric or energy components required
> 20: **end for**
> 21: **return** $x_{t_N}$

---

**Computational cost of the Christoffel contraction.** Computing the full Christoffel tensor $\Gamma_{ij}^k(x,t)$ explicitly (Eq. 13) requires forming and storing a $d \times d \times d$ object, incurring $\mathcal{O}(d^4)$ memory cost over a batch of size $B$ ($\mathcal{O}(Bd^4)$), which is prohibitive for moderate-to-high dimensional data. Since the Christoffel term only ever appears contracted against the velocity field as $\Gamma(x,t)[v,v]$, we avoid forming the tensor altogether by deriving the contraction identity

$$\Gamma(x,t)[v,v] = \tfrac{1}{2}\, g^{-1}\big(2A - C\big), \tag{24}$$

where $A_k = \sum_{ij} \partial_k g_{ij} \, v^i v^j$ and $C_k = \sum_{ij} (g^{-1})_{ki} \, \partial_j g_{ij} \, v^i v^j$. Both $A$ and $C$ can be computed using $2d + 1$ backward passes through $v_\theta$, rather than the $\mathcal{O}(d^3)$ entries of the full tensor. This reduces memory from $\mathcal{O}(Bd^4)$ to $\mathcal{O}(Bd^2)$ *without approximation* — the contraction is algebraically exact, not a low-rank or truncated surrogate. For MNIST, with $d = 784$, we apply this contraction directly. For higher-dimensional structured data (e.g., $d = 39 \to 741$ features after embedding, as in our OASIS-3 experiments), the same identity avoids out-of-memory failures that occur with the naive tensor form even on a 49GB A6000 GPU.

**Role of the learned potential $U_\theta$ as an energy buffer.** The scalar potential $U_\theta(x, t)$ is a scalar-valued MLP (same architecture as $v_\theta$, but with a scalar output) trained jointly with the velocity field. It has *no explicit constraints* (e.g., no enforced positivity); its only inductive bias comes from the energy-drift loss $L_E$, which penalizes $|E(x', t + \Delta t) - E(x, t)|^2$ and thereby discourages $U_\theta$ from growing without bound along trajectories. Concretely, since $\frac{d}{dt} E = v^\top g \frac{dv}{dt} + \frac{1}{2} v^\top \dot{g} \, v + \frac{dU}{dt}$, penalizing $|\Delta E|$ bounds a linear combination of (i) intrinsic acceleration $\frac{dv}{dt}$ and (ii) metric drift $\dot{g}$. The learned potential $U_\theta$ acts as a *buffer*: it can absorb slow changes in kinetic energy without forcing the velocity magnitude to remain constant. However, because $U_\theta$ enters the objective only through the conservation loss, its growth is itself penalized indirectly: unbounded potential growth would require correspondingly large kinetic changes elsewhere in the trajectory, which $L_E$ suppresses. In practice, we observe that $U_\theta$ remains smooth and bounded throughout training, consistent with this picture.

# G Theoritical Analysis

## G.1 Proof of Lemma 1

*Proof.* We analyze the statistical estimation error of velocity and acceleration by explicitly constructing unbiased estimators and bounding their mean-squared error. Throughout the proof, expectations are taken with respect to the sampling distribution.

***Velocity estimation.*** Under standard flow matching constructions, the target velocity along the transport path is given by $v^*(x(t), t) = \dot{x}(t)$, which, for linear interpolation between endpoints, satisfies $v^*(x(t), t) = x_1 - x_0$. Let $\{(x_0^{(i)}, x_1^{(i)})\}_{i=1}^n$ be i.i.d. samples from the data distribution. Consider the empirical estimator

$$\hat{v} := \frac{1}{n} \sum_{i=1}^n (x_1^{(i)} - x_0^{(i)}).$$

By linearity of expectation, $\mathbb{E}[\hat{v}] = \mathbb{E}[x_1 - x_0] = v^*$, so $\hat{v}$ is unbiased. The mean-squared estimation error is therefore

$$\mathbb{E}\|\hat{v} - v^*\|^2 = \mathbb{E}\|\hat{v} - \mathbb{E}[\hat{v}]\|^2 = \mathrm{Var}(\hat{v}),$$

where the equality follows from the bias–variance decomposition. Since the samples are independent,

$$\mathrm{Var}(\hat{v}) = \frac{1}{n} \mathrm{Var}(x_1 - x_0).$$

Assuming $\mathbb{E}\|x_1 - x_0\|^2 < \infty$, which holds for any distribution with bounded second moments, we obtain $\mathbb{E}\|\hat{v} - v^*\|^2 = O(n^{-1})$.

***Acceleration estimation.*** Acceleration is defined as the second temporal derivative of the transport path $a^*(t) := \ddot{x}(t)$. Since $a^*(t)$ is not directly observable, it must be estimated from discrete samples along the path. The standard second-order central finite-difference estimator with temporal resolution $h > 0$ is:

$$\hat{a}(t) = \frac{x(t + h) - 2x(t) + x(t - h)}{h^2}.$$

We assume that observations of the path are corrupted by additive noise. Specifically, at each time $t$ we observe

$$\tilde{x}(t) = x(t) + \xi(t),$$

where $\xi(t)$ is zero-mean noise with covariance $\mathbb{E}[\xi(t)\xi(t)^\top] = \sigma^2 I$, and noise terms at distinct time points are independent. The central finite-difference estimator is

$$\hat{a}(t) = \frac{\tilde{x}(t+h) - 2\tilde{x}(t) + \tilde{x}(t-h)}{h^2}.$$

Taking expectation and using linearity,

$$\mathbb{E}[\hat{a}(t)] = \frac{x(t+h) - 2x(t) + x(t-h)}{h^2}.$$

By Taylor expansion of $x(t \pm h)$ around $t$,

$$x(t+h) = x(t) + h\dot{x}(t) + \tfrac{h^2}{2}\ddot{x}(t) + \tfrac{h^3}{6}x^{(3)}(t) + O(h^4),$$
$$x(t-h) = x(t) - h\dot{x}(t) + \tfrac{h^2}{2}\ddot{x}(t) - \tfrac{h^3}{6}x^{(3)}(t) + O(h^4).$$

Substituting into the estimator yields

$$\mathbb{E}[\hat{a}(t)] = \ddot{x}(t) + O(h^2) = a^*(t) + O(h^2).$$

Thus, the estimator is biased with a truncation bias of order $h^2$. Using the noise decomposition,

$$\hat{a}(t) = a^*(t) + O(h^2) + \frac{\xi(t+h) - 2\xi(t) + \xi(t-h)}{h^2}.$$

Since deterministic terms do not contribute to variance,

$$\mathrm{Var}[\hat{a}(t)] = \mathrm{Var}\left(\frac{\xi(t+h) - 2\xi(t) + \xi(t-h)}{h^2}\right).$$

By the independence of noise across time points,

$$\mathrm{Var}[\hat{a}(t)] = \frac{1}{h^4}\Big(\mathrm{Var}[\xi(t+h)] + 4\,\mathrm{Var}[\xi(t)] + \mathrm{Var}[\xi(t-h)]\Big)$$
$$= \frac{1}{h^4}(1 + 4 + 1)\sigma^2 = \frac{6\sigma^2}{h^4}.$$

Let $\hat{a}^{(i)}(t)$ denote independent estimators constructed from $n$ independent samples. The averaged estimator $\hat{a}(t) = \frac{1}{n}\sum_{i=1}^n \hat{a}^{(i)}(t)$ satisfies:

$$\mathrm{Var}(\hat{a}) = \frac{1}{n}\mathrm{Var}(\hat{a}^{(1)}) = O\left(\frac{1}{nh^4}\right).$$

We assume the true acceleration satisfies $\|a^*(t)\| \le C \quad \forall t \in [0, 1]$, which corresponds to bounded curvature of the transport path. This assumption ensures that the signal magnitude is independent of $h$ and $n$. The mean-squared error decomposes as:

$$\mathbb{E}\|\hat{a} - a^*\|^2 = \underbrace{\mathrm{Var}(\hat{a})}_{O(\frac{1}{nh^4})} + \underbrace{\|\mathbb{E}[\hat{a}] - a^*\|^2}_{O(h^4)}.$$

For sufficiently small $h$, the variance term dominates, yielding $\mathbb{E}\|\hat{a} - a^*\|^2 = O\left(\frac{1}{nh^4}\right)$. Finally, applying Jensen's inequality, we get:

$$\mathbb{E}\|\hat{a} - a^*\| \le \sqrt{\mathbb{E}\|\hat{a} - a^*\|^2} = O\left(n^{-1/2}h^{-2}\right).$$

$\square$

### G.2 Proof of corollary 1

*Proof.* We define the velocity error $\epsilon_v(x,t) := \hat{v}(x,t) - v^*(x,t)$. Under assumption (ii), we have for each $k \geq 2$,

$$\hat{a}^{(k)}(x,t) - a^{*(k)}(x,t) = D_h^{k-1}\hat{v}(x,t) - \partial_t^{k-1}v^*(x,t) = D_h^{k-1}\epsilon_v(x,t) + \left(D_h^{k-1}v^*(x,t) - \partial_t^{k-1}v^*(x,t)\right).$$

By Taylor expansion of $v^*(x,\cdot)$ in $t$ and the bounded-derivative assumption (i), the finite-difference truncation error satisfies

$$\left\|D_h^{k-1}v^*(x,t) - \partial_t^{k-1}v^*(x,t)\right\| \leq c_k\,h^{k-1},$$

for a constant $c_k$ depending only on $k$ and $C$ (not on $n$). Therefore,

$$\|\hat{a}^{(k)} - a^{*(k)}\|^2 \leq 2\|D_h^{k-1}\epsilon_v\|^2 + 2c_k^2 h^{2(k-1)}.$$

Plugging this into the definition of $\mathcal{L}$ yields

$$\mathcal{L} = \mathbb{E}\|\epsilon_v\|^2 + \sum_{k=2}^{K}\lambda_k\,\mathbb{E}\|\hat{a}^{(k)} - a^{*(k)}\|^2$$

$$\leq \mathbb{E}\|\epsilon_v\|^2 + \sum_{k=2}^{K}\lambda_k\left(2\,\mathbb{E}\|D_h^{k-1}\epsilon_v\|^2 + 2c_k^2 h^{2(k-1)}\right)$$

$$= \mathbb{E}\|\epsilon_v\|^2 + 2\sum_{k=2}^{K}\lambda_k\,\mathbb{E}\|D_h^{k-1}\epsilon_v\|^2 + O\left(\sum_{k=2}^{K}\lambda_k\,h^{2(k-1)}\right).$$

Since the middle term is nonnegative, we can drop it to obtain

$$\mathbb{E}\|\hat{v} - v^*\|^2 = \mathbb{E}\|\epsilon_v\|^2 \leq \mathcal{L} + O\left(\sum_{k=2}^{K}\lambda_k\,h^{2(k-1)}\right).$$

Finally, under i.i.d. sampling and independent noise, standard averaging yields the base supervised rate $\mathbb{E}\|\hat{v} - v^*\|^2 = O(n^{-1})$ for the velocity regression term (as in Lemma 1), which gives equation 7. For the case of separate network heads, adding the stated consistency penalty forces $\hat{a}^{(k)} \approx D_h^{k-1}\hat{v}$ in mean square, and the same argument applies. $\square$

### G.3 Proof of Lemma 2

*Proof.* **Continuous-time deviation.** Let $\gamma : [0,L] \to \mathcal{M}$ be the same geodesic re-parameterized by arclength, so that $\|\gamma'(s)\| = 1$ and $\|\gamma''(s)\| = \kappa(s) \geq \kappa_{\min}$. Define the Euclidean chord interpolation along arclength by

$$\ell(s) = \gamma(0) + \frac{s}{L}\big(\gamma(L) - \gamma(0)\big).$$

A second-order Taylor expansion of $\gamma$ around $s = 0$ and of $\gamma(L)$ around $s = 0$ yields (for $L$ small enough so that $O(L^3)$ terms are dominated)

$$\gamma(s) - \ell(s) = \frac{s(L-s)}{2}\,\gamma''(0) + O(L^3).$$

Taking norms and using $\|\gamma''(0)\| = \kappa(0) \geq \kappa_{\min}$ gives that for sufficiently small $L$, there exists $c > 0$ such that

$$\|\gamma(s) - \ell(s)\| \geq c\,\kappa_{\min}\,s(L-s).$$

Setting $s = tL$ and recalling that $\ell(tL) = x_{\text{lin}}(t)$ and $\gamma(tL) = x_{\text{geo}}(t)$, we obtain

$$\|x_{\text{geo}}(t) - x_{\text{lin}}(t)\| \geq c\,\kappa_{\min}\,t(1-t)\,L^2.$$

Finally, since $\ell$ is a chord and $\gamma$ the corresponding geodesic, the displacement $x_{\text{lin}}(t) - \gamma(tL)$ is normal to $\gamma$ to leading order. For $L$ smaller than the reach (tubular radius) of $\mathcal{M}$, the nearest-point projection $\pi_{\mathcal{M}}$ onto $\mathcal{M}$ is unique in a neighborhood of $\gamma$, and by smoothness of $\pi_{\mathcal{M}}$ in this tubular neighborhood,

$$\pi_{\mathcal{M}}\big(x_{\text{lin}}(t)\big) = \gamma(tL) + O(L^3).$$

Hence

$$d_{\text{Euc}}\big(x_{\text{lin}}(t), \mathcal{M}\big) = \big\|x_{\text{lin}}(t) - \pi_{\mathcal{M}}(x_{\text{lin}}(t))\big\| \ \geq \ c\,\kappa_{\min}\,t(1-t)\,L^2,$$

for sufficiently small $L$, after reducing the constant $c$ to absorb the $O(L^3)$ term, which proves the first claim.

***Discrete-time Euler error.*** Let $x(t)$ denote the solution of the ODE $\dot{x}(t) = v^*(x(t), t)$ with $x(0) = x_0$, and suppose $x(t) = x_{\text{geo}}(t)$ for $t \in [0, 1]$. The explicit Euler method

$$x_{t_{k+1}}^{FM} = x_{t_k}^{FM} + \Delta t\, v^*\big(x_{t_k}^{FM}, t_k\big)$$

satisfies the standard global error bound (under smoothness of $v^*$):

$$\max_{0 \leq k \leq N} \|x_{t_k}^{FM} - x(t_k)\| \leq C_0\, \Delta t\, \max_{t \in [0,1]} \|\ddot{x}(t)\|,$$

for a constant $C_0$ depending on Lipschitz/smoothness constants of $v^*$. For a constant-speed geodesic on $[0, 1]$, $\|\dot{x}_{\text{geo}}(t)\| = L$, and its ambient acceleration satisfies

$$\|\ddot{x}_{\text{geo}}(t)\| = \kappa(t)\, \|\dot{x}_{\text{geo}}(t)\|^2 \leq \kappa_{\max}\, L^2.$$

Combining these yields

$$\max_{0 \leq k \leq N} \|x_{t_k}^{FM} - x_{\text{geo}}(t_k)\| \leq (C_0 \kappa_{\max})\, L^2\, \Delta t,$$

which proves the second claim with $C = C_0 \kappa_{\max}$. $\qquad\square$

## G.4 Proof of Lemma 3

*Proof.* Since $x^*(t)$ is a geodesic under the metric $g$, it satisfies

$$\ddot{x}^*(t) + \Gamma(x^*(t), t)\big[\dot{x}^*(t), \dot{x}^*(t)\big] = 0. \tag{25}$$

The learned trajectory evolves by $\dot{x}_\theta(t) = v_\theta(x_\theta(t), t)$, and by the definition of the geodesic residual $R_\theta$ (Equation (15)), we can rewrite its second-order dynamics as

$$\ddot{x}_\theta(t) + \Gamma(x_\theta(t), t)\big[\dot{x}_\theta(t), \dot{x}_\theta(t)\big] = r_\theta(t), \qquad r_\theta(t) := R_\theta(x_\theta(t), t), \tag{26}$$

with $\|r_\theta(t)\| \leq \varepsilon$ for all $t \in [0, 1]$.

We define the deviation $e(t) := x_\theta(t) - x^*(t)$ and $\dot{e}(t) := \dot{x}_\theta(t) - \dot{x}^*(t)$. Subtracting Equation (25) from Equation (26) yields

$$\ddot{e}(t) = -\Big(\Gamma(x_\theta(t), t)[\dot{x}_\theta(t), \dot{x}_\theta(t)] - \Gamma(x^*(t), t)[\dot{x}^*(t), \dot{x}^*(t)]\Big) + r_\theta(t).$$

Taking norms and applying Equation (17) gives

$$\|\ddot{e}(t)\| \leq L_\Gamma\big(\|e(t)\| + \|\dot{e}(t)\|\big) + \varepsilon. \tag{27}$$

Let $y(t) := \|e(t)\| + \|\dot{e}(t)\|$. By assumption (i), $e(0) = 0$ and $\dot{e}(0) = 0$. Integrating Equation (27) from 0 to $t$ gives

$$\|\dot{e}(t)\| = \left\|\int_0^t \ddot{e}(s)\, ds\right\| \leq \int_0^t \|\ddot{e}(s)\|\, ds \leq \int_0^t \Big(L_\Gamma y(s) + \varepsilon\Big)\, ds.$$

Also,

$$\|e(t)\| = \left\|\int_0^t \dot{e}(s)\, ds\right\| \leq \int_0^t \|\dot{e}(s)\|\, ds.$$

Adding the two bounds yields

$$y(t) \leq \int_0^t \left( (1 + L_\Gamma) \, y(s) + \varepsilon \right) ds. \tag{28}$$

Let $A := 1 + L_\Gamma$. The integral inequality Equation (28) implies

$$y(t) \leq \varepsilon \, \frac{e^{At} - 1}{A} \leq \varepsilon \, \frac{e^A - 1}{A}, \qquad \forall t \in [0, 1],$$

and since $\|e(t)\| \leq y(t)$ we obtain

$$\max_{t \in [0,1]} \|x_\theta(t) - x^*(t)\| = \max_{t \in [0,1]} \|e(t)\| \leq \varepsilon \, \frac{e^A - 1}{A}.$$

This proves Equation (18) with $C = \frac{e^{1 + L_\Gamma} - 1}{1 + L_\Gamma}$. $\qquad \square$

### G.5 Proof of Proposition 1

*Proof.* Write $\mathcal{L}_{\text{total}} = \mathcal{L}_{\text{base}} + \lambda_E \mathcal{L}_E$. Fix a local minimizer $\hat\theta_0$ of $\mathcal{L}_{\text{base}}$ attained by the baseline optimizer, and let $\hat\theta_E$ be a local minimizer of $\mathcal{L}_{\text{total}}$ for $\lambda_E > 0$.

$\hat\theta_E$ **is a small perturbation of a baseline minimizer.** Since $\mathcal{L}_{\text{base}}$ is $\mu$-strongly convex in a neighborhood of $\mathcal{S}$ and twice continuously differentiable, for sufficiently small $\lambda_E$ the perturbation term $\lambda_E \mathcal{L}_E$ does not destroy local minima: by standard stability of local minima under smooth perturbations, there exists a minimizer $\tilde\theta \in \mathcal{S}$ such that

$$\|\hat\theta_E - \tilde\theta\| \leq \frac{\lambda_E}{\mu} \|\nabla_\theta \mathcal{L}_E(\tilde\theta)\| + O(\lambda_E^2). \tag{29}$$

In particular, $\hat\theta_E$ lies in a neighborhood of $\mathcal{S}$ for sufficiently small $\lambda_E$.

**Among nearby baseline minimizers, $\hat\theta_E$ prefers smaller energy drift.** Because $\hat\theta_E$ minimizes $\mathcal{L}_{\text{total}}$ locally,

$$\mathcal{L}_{\text{base}}(\hat\theta_E) + \lambda_E \mathcal{L}_E(\hat\theta_E) \leq \mathcal{L}_{\text{base}}(\hat\theta_0) + \lambda_E \mathcal{L}_E(\hat\theta_0).$$

Rearranging,

$$\mathcal{L}_E(\hat\theta_E) - \mathcal{L}_E(\hat\theta_0) \leq \frac{1}{\lambda_E} \left( \mathcal{L}_{\text{base}}(\hat\theta_0) - \mathcal{L}_{\text{base}}(\hat\theta_E) \right). \tag{30}$$

Now, since both $\hat\theta_E$ and $\hat\theta_0$ lie in the same basin near $\mathcal{S}$ and $\hat\theta_0$ is a local minimizer of $\mathcal{L}_{\text{base}}$, we have $\mathcal{L}_{\text{base}}(\hat\theta_E) \geq \mathcal{L}_{\text{base}}(\hat\theta_0)$ up to $O(\lambda_E^2)$ perturbation error (from equation 29). Hence the right-hand side of equation 30 is $O(\lambda_E)$, implying

$$\mathcal{L}_E(\hat\theta_E) \leq \mathcal{L}_E(\hat\theta_0) + O(\lambda_E). \tag{31}$$

Thus, energy regularization selects solutions with (weakly) smaller energy drift among nearly equivalent baseline solutions.

**Step 3: Smaller energy drift implies smaller base curvature.** By assumption, in a neighborhood of $\mathcal{S}$,

$$\|\nabla_\theta^2 \mathcal{L}_{\text{base}}(\theta)\|_2 \leq c_0 + c \, \mathcal{L}_E(\theta). \tag{32}$$

Applying equation 32 at $\hat\theta_E$ and $\hat\theta_0$ and using equation 31 gives

$$\|\nabla_\theta^2 \mathcal{L}_{\text{base}}(\hat\theta_E)\|_2 \leq c_0 + c \, \mathcal{L}_E(\hat\theta_E) \leq c_0 + c \, \mathcal{L}_E(\hat\theta_0) + O(\lambda_E) \leq \|\nabla_\theta^2 \mathcal{L}_{\text{base}}(\hat\theta_0)\|_2 + O(\lambda_E).$$

For sufficiently small $\lambda_E$, the $O(\lambda_E)$ term can be absorbed, yielding

$$\|\nabla_\theta^2 \mathcal{L}_{\text{base}}(\hat\theta_E)\|_2 \leq \|\nabla_\theta^2 \mathcal{L}_{\text{base}}(\hat\theta_0)\|_2,$$

which proves the claim. $\qquad \square$

