# OpenReview forum: "GE-FM: Geometry-aware Energy-based Flow Matching for Non Euclidean Manifolds"
_TMLR — Accepted by TMLR_

### Review · Reviewer_Q2v3 · 2026-04-16

**Summary Of Contributions:**

## Summary

This paper studies the limitations of flow matching in the few-step sampling regime, particularly when the target distribution lies on a highly curved data manifold. The authors argue that standard Euclidean flow matching suffers from a mismatch between the learned transport dynamics and the underlying geometry, while explicit higher-order modeling may introduce instability and discretization sensitivity.

To address this issue, the paper proposes GE-FM, a geometry-aware and energy-based first-order flow matching framework. The method constructs an induced metric from the velocity Jacobian, introduces Christoffel-symbol-based corrections to account for curvature, and imposes approximate energy conservation to stabilize both training and sampling. Empirically, the method demonstrates promising improvements on synthetic curved datasets and also reports one-step generation results on MNIST.

Overall, the paper is interesting and well motivated. The core problem is meaningful, the proposed perspective is novel, and the method is conceptually coherent. That said, the current empirical validation remains somewhat limited, and both the computational and theoretical aspects would benefit from further clarification.

## Strengths

### 1. Clear motivation and well-defined problem setting

The paper identifies a meaningful limitation of existing flow matching methods in the few-step regime, especially when the underlying transport trajectories are highly curved. The distinction between geometric mismatch in first-order Euclidean formulations and the  instability of explicit higher-order modeling is clearly articulated and convincing.

### 2. Methodologically interesting and conceptually coherent

The proposed framework combines geometry-aware transport modeling  with energy-based stabilization in a unified and reasonably elegant way. The use of an induced metric, Christoffel corrections, and approximate energy conservation is conceptually appealing, and the overall design feels coherent rather than ad hoc.

### 3. Good intuition and generally strong presentation

The paper is well organized and relatively easy to follow. In particular, the theoretical discussion provides useful intuition for why the proposed method could improve few-step transport on curved manifolds. While not every claim is fully rigorous, the presentation does a good job of motivating the method and explaining its intended benefits.



## Weaknesses

### 1. The experimental evaluation is still relatively limited

The current empirical section feels somewhat narrow. Most of the evidence comes from synthetic high-curvature datasets, which are useful for illustrating the core intuition behind the method, but are not sufficient to demonstrate that the approach would remain effective on more realistic or large-scale generative modeling tasks.
The MNIST experiment is a useful addition, but by current standards it is still a fairly lightweight benchmark. To make the empirical case substantially stronger, the paper should include experiments on more standard image-generation datasets and under more broadly accepted few-step evaluation settings.

### 2. Computational cost and scalability are under-discussed

One important missing piece is a careful discussion of computational overhead. Compared with standard flow matching, GE-FM appears meaningfully more expensive, since it introduces Jacobian-based metric construction, Christoffel-symbol computation, and an additional potential/energy-related term.
However, the paper does not sufficiently quantify the practical cost of these components in terms of runtime, memory usage, numerical stability, and scalability to higher-dimensional problems. This makes it difficult to assess the practical value of the method beyond its conceptual appeal and reported sample quality improvements.

**Audience:**

Yes

**Audience Explanation:**

Because the paper provides sufficiently clear and convincing evidence for its main claim. The experiments on curved synthetic datasets are well matched to the paper’s core motivation and support the argument that the proposed geometry-aware formulation improves few-step flow matching in high-curvature settings.

However, the evidence is more limited for the paper’s broader claims. The empirical validation is still relatively narrow, and the MNIST experiment alone is not enough to establish effectiveness on more realistic large-scale generative modeling benchmarks. In addition, the paper would benefit from a clearer discussion of computational overhead, scalability, and the scope of the theoretical conclusions.

Overall, I believe the central claim is supported, but some broader implications would require stronger empirical and theoretical backing.

**Broader Impact Concerns:**

No impact concerns.

**Claims And Evidence:**

Yes

**Claims Explanation:**

The paper provides evidence that is generally clear and convincing for its central claim, namely that geometry-aware corrections can improve few-step flow matching when the transport paths are highly curved. The experiments are well targeted to this question, and the empirical results are consistent with the proposed intuition.

**Requested Changes:**

### 1. Strengthen the empirical evaluation

The paper would benefit from a broader empirical study beyond synthetic curved datasets and MNIST.

### 2. Add a clear analysis of computational overhead

The paper should provide a more explicit discussion, and ideally quantitative measurements, of the additional computational cost introduced by GE-FM. This includes runtime, memory consumption, and any implementation or stability challenges associated with the Jacobian-based metric, Christoffel-symbol corrections, and energy regularization. Such analysis is important for understanding the practical trade-off between performance gains and computational complexity.

---

> ### Author Response · Authors · 2026-05-15
> **Response (part 1)**
>
> We thank the reviewer for identifying the work as an important step towards solving a key **"limitation of flow matching models"** and for appreciating the **"unified and elegant way"** of combining geometry-aware transport modeling and energy-based stabilization. We also thank the reviewer for acknowledging **"the paper is interesting and well motivated"** and highlighting the **meaningful problem formulation**, **novel proposed perspective**, and **"the method is conceptually coherent"**.
>
> *We break the response in two parts due to character limit of each response.*
>
> **@ Analysis of computational overhead:**
>
> **Table 1: Training and test times (seconds) for 1-step setting.**
>
> | Dataset | Method | Version | Train | Test |
> |---|---|---|---|---|
> | Irr. Circle | FM | -- | 28.95 | 0.65 |
> | Irr. Circle | SC | -- | 31.02 | 0.71 |
> | Irr. Circle | GE-FM (Energy optimization) | w/o batch reordering | 421.46 | 0.88 |
> | Irr. Circle | GE-FM (Energy optimization) | w batch reordering | 1382.85 | 0.80 |
> | Irr. Circle | GE-FM (Christoffel correction) | w/o batch reordering | 530.89 | 0.67 |
> | Irr. Circle | GE-FM (Christoffel correction) | w batch reordering | 1549.45 | 0.79 |
> | Irr. Circle | GE-FM | w/o batch reordering | 525.18 | 0.87 |
> | Irr. Circle | GE-FM | w batch reordering | 1604.65 | 0.82 |
> | 3-spin Spiral | FM | -- | 28.37 | 0.67 |
> | 3-spin Spiral | SC | -- | 26.56 | 0.65 |
> | 3-spin Spiral | GE-FM (Energy optimization) | w/o batch reordering | 406.15 | 0.88 |
> | 3-spin Spiral | GE-FM (Energy optimization) | w batch reordering | 1112.47 | 0.83 |
> | 3-spin Spiral | GE-FM (Christoffel correction) | w/o batch reordering | 551.48 | 0.94 |
> | 3-spin Spiral | GE-FM (Christoffel correction) | w batch reordering | 1275.62 | 0.81 |
> | 3-spin Spiral | GE-FM | w/o batch reordering | 503.27 | 0.91 |
> | 3-spin Spiral | GE-FM | w batch reordering | 1372.39 | 0.84 |
>
> We report runtimes for the 1-step setting on two curved benchmarks in Table 1. GE-FM's induces only additional training cost, while inference remains close to baselines (<1s). The geometry-aware and energy-based components introduce extra computations during optimization, whereas inference still uses a first-order transport model. Moreover, batch reordering further increases training time due to explicit matching/reassignment, but offers a small speedup at inference. This is a direct consequence of improved source–target alignment during training, helping the model converge to a smoother transport map. ***Trade-off: increased training overhead in exchange for improved geometric fidelity and transport quality, with negligible deployment-time cost.***
>
> **Table 2: $d_{\min}$ for 1-step setting comparing the components of GE-FM.**
>
> | Dataset | Method | $d_{\min}$ |
> |---|---|---|
> | Irr. Circle | GE-FM (Energy optimization) | 6.49 |
> | Irr. Circle | GE-FM (Christoffel correction) | 6.42 |
> | Irr. Circle | GE-FM | 6.35 |
> | 3-Spin Spiral | GE-FM (Energy optimization) | 51.75 |
> | 3-Spin Spiral | GE-FM (Christoffel correction) | 53.10 |
> | 3-Spin Spiral | GE-FM | 49.75 |
>
> Table 2 isolates the contributions of GE-FM's two components. On both datasets, the full model achieves the best $d_{\min}$, showing that energy-based stabilization and Christoffel correction are complementary. The former improves large-step stability, while the latter enhances geometric alignment with curved trajectories. Our results on the challenging 3-spiral benchmark, which is highly curved, validates our claim. Batch reordering (Table 1, main paper) further improves performance as denoising steps increase, owing to better source–target alignment.

---

> > ### Author Response · Authors · 2026-05-15
> > **Response (part 2)**
> >
> > **@ Strengthening the empirical evaluation:** To demonstrate that our technique extends beyond synthetic datasets to real-world problems, we evaluate on resting-state fMRI data. The task is class-conditional generation of functional connectivity matrices—Symmetric Positive Definite (SPD) covariance matrices estimated from fMRI time series over 39 brain regions, conditioned on diagnosis (healthy vs Alzheimer's). This challenging real-world setting combines strong non-linearity with the data scarcity typical of medical imaging. For this, we have identified OASIS-3 (https://www.medrxiv.org/content/10.1101/2019.12.13.19014902v1) as a standard benchmark.
> >
> > We follow the protocol of DiffeoCFM (https://arxiv.org/pdf/2505.18193): MSDL atlas parcellation, 200 training epochs, corrcholesky diffeomorphism embedding, and 10-fold GroupShuffleSplit cross-validation with subject-disjoint splits *to ensure a fair, apples-to-apples comparison*. Each SPD matrix is embedded into Euclidean space via the corrcholesky diffeomorphism (a bijection to the strictly lower-triangular Cholesky factor), standardized, and passed to the generative model. For generation, outputs are mapped back to the SPD manifold via the inverse diffeomorphism. We report $\alpha$-Precision and $\beta$-Recall (fidelity and diversity of the generated distribution), their harmonic mean $\alpha,\beta$-F1, and CAS metrics (ROC-AUC and F1), which measure whether a classifier trained solely on synthetic data generalizes to held-out real data, directly probing downstream utility. All metrics are averaged over 10 splits (mean ± std).
> >
> > **Table 3: Quantitative comparison on OASIS-3.**
> >
> > | Method | $\alpha$-Precision ↑ | $\beta$-Recall ↑ | $\alpha,\beta$-F1 ↑ | ROC-AUC ↑ | F1 ↑ |
> > |---|---|---|---|---|---|
> > | Real Data | 0.89 ± 0.04 | 0.88 ± 0.05 | 0.88 ± 0.01 | 0.71 ± 0.04 | 0.61 ± 0.06 |
> > | TriangCFM | 0.22 ± 0.27 | 0.26 ± 0.37 | 0.23 ± 0.32 | 0.51 ± 0.05 | 0.45 ± 0.17 |
> > | DiffeoGauss | 0.52 ± 0.06 | 0.31 ± 0.02 | 0.39 ± 0.03 | 0.68 ± 0.03 | 0.40 ± 0.09 |
> > | DiffeoCFM | 0.63 ± 0.06 | 0.38 ± 0.04 | 0.47 ± 0.05 | 0.68 ± 0.03 | 0.53 ± 0.04 |
> > | **GE-FM** | **0.55 ± 0.01** | **0.42 ± 0.01** | **0.48 ± 0.01** | **0.71 ± 0.04** | **0.55 ± 0.07** |
> >
> > In Table 3, we have: (i) *Real Data* (oracle), where a classifier is trained and tested on real data (ii) *TriangCFM*, a geometry-agnostic flow matching baseline in the lower-triangular embedding (iii) *DiffeoGauss*, a class-conditional Gaussian in the corrcholesky space capturing first- and second-order statistics (iv) *DiffeoCFM*, manifold-aware conditional flow matching. *GE-FM requires no diffeomorphic preprocessing*. It augments standard CFM with a *manifold-aware yet agnostic* penalty induced by $g$, and corrects sampling via $-\Gamma[v,v]$, encouraging geodesic-consistent transport. For fair comparison, we use the same corrcholesky embedding as DiffeoCFM. To address $\mathcal{O}(d^4)$ cost of the exact Christoffel tensor, we derive a contraction $\Gamma[v,v] = \frac{1}{2}g^{-1}(2A - C)$, reducing computation to $2d+1$ backward passes and memory ($\mathcal{O}(Bd^4)$ $\Rightarrow$ $\mathcal{O}(Bd^2)$) without approximation, critical for high-dimensional embeddings ($d=39 \Rightarrow 741$ features), where the naive form causes OOM even on a 49GB A6000 GPU.
> >
> > On OASIS-3, GE-FM improves $\alpha,\beta$-F1 over DiffeoCFM with ~5× lower variance, indicating more stable generation, and achieves the best ROC-AUC (matching Real Data) while improving CAS-F1. The trade-off is lower $\alpha$-Precision, as Christoffel correction spreads mass more uniformly along the manifold, boosting $\beta$-Recall at the cost of slight deviations from high-density regions. Thus, DiffeoCFM is sharper but mode-seeking, whereas GE-FM provides broader, more representative coverage, with improved CAS metrics confirming the added diversity is meaningful. TriangCFM collapses, underscoring the need for geometry-awareness, while DiffeoGauss, though competitive in ROC-AUC, suffers low recall, reflecting the limits of unimodal assumptions for multimodal neuroimaging data.

---

### Review · Reviewer_5TGe · 2026-04-24

**Summary Of Contributions:**

This paper proposes GE-FM, a geometry-aware flow matching method that is strictly first-order and respects the curved and time-evolving geometry of diffusion manifolds instead of assuming Euclidean geometry. In flow matching or neural differential equations, the model learns a velocity field over which the latent vectors evolve over time by integrating over the velocity field. GE-FM defines a total energy as the sum of kinetic energy and potential energy, with the former computed from the predicted velocity field and the latter directly learned, and enforces approximate energy conservation along transport trajectories.

The authors claim that the energy formulation induces a time-dependent Riemannian metric that captures the evolving diffusion geometry without explicit manifold supervision. They further use Christoffel-symbol correction to adjust the velocity field to account for curvature and implicitly model the higher-order effects using a first-order formulation. This Christoff correction does not assume the form of the manifold and can automatically adapt to the evolving diffusion geometry.

Theoretically, the authors made the following claims: (1) higher-order supervision is noisy, (2) Euclidean flow matching suffers from curvature-induced bias on non-Euclidean manifolds. With these shown, GE-FM is introduced as a unified remedy to both issues.
Empirically, the authors evaluated GE-FM on at least five synthetic datasets and a real-world MNIST dataset to show superior performance than flow matching baselines.

**Audience:**

Yes

**Audience Explanation:**

The topic of improving flow matching methods is highly interesting and relevant.

**Broader Impact Concerns:**

No concerns.

**Claims And Evidence:**

Yes

**Claims Explanation:**

The motivation and rationale are nicely explained. The theoretical results add to the background, and the empirical results demonstrate some improvement over the baseline. With that said, the empirical performance do not look good enough, which I will explain in the Weaknesses section below.

Strengths

1. The technical story is very clear. The energy formulation and Christoffel-symbol correction are presented as well-motivated remedies to well-discussed failure modes of standard flow matching.
2. In Figure 1, the illustrations of GE-FM as well as the Christoffel-symbol correction and energy-based optimization are very nice.
3. In Figure 2, the loss landscape smoothness is very intuitive.
4. In figure 3a, the ablation of Christoffel-symbol correction shows visually obvious improvement induced by the component.

Weaknesses
1. While the loss landscape and the ablation on Christoffel-symbol correction look decent, the other empirical results do not look very persuasive. I will elaborate below.

    1.1. In Table 1, I am not sure if $d_\text{min}$ is the most representative metric for evaluating flow matching tasks. In image generation, FID would be standard, while for point clouds, I thought it would be something along the lines of Earth Mover’s distance.

    1.2 The performance even on the synthetic dataset does not look very promising. For example, in Figure 3a and Figure 4, even though GE-FM shows cleaner and sharper generated distribution, it does not nicely cover the entire spiral distribution. The generated points tend to collapse to the inner spiral while leaving the outer spiral not covered. It might be because the baseline flow matching method or implementation used in this submission is not good enough.

2. It might improve the empirical contribution of the paper if the authors can include the comparisons against a few more flow matching methods such as conditional flow matching (CFM) [1], OT-CFM [1], curly flow matching [2], meta flow matching [3]. Note these are just some nominations and some of them might not be the most appropriate for comparison. With that said, some of them might have better implementation which GE-FM can build on top for better empirical performance. Besides, it would be nice include experiments on one more real-world dataset.

[1] Tong, Alexander, et al. "Improving and generalizing flow-based generative models with minibatch optimal transport." TMLR 2024.

[2] Petrović, Katarina, et al. "Curly Flow Matching for Learning Non-gradient Field Dynamics." NeurIPS 2025.

[3] Atanackovic, Lazar, et al. "Meta Flow Matching: Integrating Vector Fields on the Wasserstein Manifold." ICLR 2025.

**Requested Changes:**

My main concern is the empirical weakness as I described in Weaknesses #1-2.

Some other minor things include
1. I would appreciate a quantitative ablation study of the 3 loss terms introduced in Equation 22.
2. Minor overleaf issue: many citations in this paper should be inserted using `\citep{}` instead of `\cite{}`.

---

> ### Author Response · Authors · 2026-05-26
> **Response (Part 1)**
>
> We thank the reviewer for highlighting that **the technical story is clear and well-motivated** and for appreciating the **presentation of the energy formulation and Christoffel-symbol correction as principled remedies to the identified failure modes**. We also thank the reviewer for recognizing the effectiveness of the visual illustrations (Figure 1), the loss landscape analysis (Figure 2), and the ablation demonstrating the impact of Christoffel correction (Figure 3a), which collectively support the geometric intuition behind the proposed method.
>
> *We break the response into two parts to be consistent with the character limit of each response.*
>
> **@ Choice of evaluation metrics:** We agree that FID (for images) and Earth Mover's Distance (EMD) (for point sets) are standard evaluation metrics in generative modeling literature. However, our objective is to evaluate *transport quality under curvature*, which requires explicitly measuring how well generated samples remain aligned with the underlying data manifold. For this reason, we adopt $d_{\min}$, which directly measures the Euclidean distance between generated samples and the target manifold and is also used in prior work studying geometric transport behavior in similar settings (https://arxiv.org/pdf/2502.00688v1). This metric is particularly sensitive to *off-manifold drift*, which is the core failure mode identified in Lemma 2. Thus, $d_{\min}$ is specifically designed to measure transport fidelity on curved manifolds, as it quantifies the minimum distance between generated samples and the target manifold, directly capturing off-manifold drift that standard metrics like FID do not penalize. In this sense, $d_{\min}$ serves as a proxy for geodesic alignment, since lower values indicate trajectories that better follow the intrinsic manifold geometry rather than shortcutting through ambient Euclidean space. ***While EMD captures global distributional alignment, it does not explicitly penalize deviations from the intrinsic manifold geometry.*** This is primarily due to EMD's ground metric being typically defined in the ambient Euclidean space. In highly curved settings (e.g., spirals), two distributions can have low EMD while still differing significantly in how well they follow the manifold structure (e.g., cutting across low-density regions).
>
> To complement $d_{\min}$, we additionally evaluate on a real-world highly non-linear setting, i.e., OASIS-3 benchmark (please see response to Reviewer Q2v3, Table 3), where we report $\alpha$-Precision, $\beta$-Recall, $\alpha\beta$-F1, and downstream classification metrics (ROC-AUC, F1). These metrics collectively measure fidelity, diversity, and task-level utility of generated samples. The results demonstrate that GE-FM not only improves geometric alignment (captured by $d_{\min}$) but also achieves strong performance across standard generative modeling metrics, indicating that the benefits extend beyond synthetic curvature-focused settings.
>
> **@ Partial coverage of spiral distributions:** *Great observation!* We emphasize that this behavior is *not unique to GE-FM*, but is a known challenge in few-step flow matching, where large step sizes induce mode-averaging and trajectory collapse. As shown in Figure 3a, GE-FM significantly improves geometric alignment compared to standard flow matching by reducing off-manifold drift and producing sharper trajectories. However, full coverage of highly curved, multi-turn manifolds in the *one-step regime* remains inherently difficult due to the limited expressivity of few-step transport. Importantly, Table 1 of the main paper shows that coverage improves consistently as the number of denoising steps increases, indicating that the proposed method mitigates (but does not entirely eliminate) this fundamental limitation. *If the reviewer suggests, we will add a discussion to clarify this point explicitly in the paper*.

---

> ### Author Response · Authors · 2026-05-26
> **Response (Part 2)**
>
> **@ Additional baselines:** i) CFM and OT-CFM [1] improve minibatch coupling quality via optimal transport to straighten trajectories, but they remain fundamentally Euclidean first-order methods and do not introduce any geometry-aware correction. Since the failure mode we target — curvature-induced bias on highly curved manifolds — is orthogonal to coupling quality, these methods do not constitute the most informative comparison for our specific claims. That said, our energy-based batch reordering (Appendix A.2) is conceptually related to OT-CFM's minibatch coupling, and we discuss this connection explicitly therein. Curly Flow Matching [2] is designed to model non-gradient (curl-containing) vector fields for learning non-conservative dynamics, which is a complementary objective to ours. GE-FM targets curvature bias and dynamical instability in few-step regimes, not the non-gradient structure of the field itself.
>
> ii) Meta Flow Matching [3] operates on the Wasserstein manifold to integrate population-level dynamics, a fundamentally different problem setting from single-distribution transport on curved data manifolds.
>
> In contrast, DiffeoCFM is directly designed for geometry-aware flow matching on non-Euclidean spaces and serves as the most meaningful comparison. We additionally include Metric Flow Matching (MFM), *suggested by Reviewer bt4b*, as it also learns a data-dependent Riemannian metric for flow matching and is the most structurally similar prior work to GE-FM. *As shown in Table 1 of Response (Part 3) of Reviewer bt4b*, GE-FM outperforms both DiffeoCFM and MFM across all metrics on OASIS-3, with notably tighter variance.
>
> **@ Additional empirical validation:** *Nice suggestion!* As also noted by Reviewer Q2v3, our experiments are intentionally centered on highly curved synthetic datasets to isolate the geometric failure mode in a controlled setting. To complement this, in addition to the synthetic highly curved setups and MNIST, we add experimental results on OASIS-3 (please see response to Reviewer Q2v3, Table 3) for a functional brain connectivity task, which is a highly non-linear real-world medical imaging task with an additional constraint of limited data for training the generative pipeline (https://arxiv.org/pdf/2505.18193). Downstream performance on the generated samples also validates that GE-FM produces valid and interpretable samples due to its geometric consistency and prevents hallucinated or clinically irrelevant samples.
>
> **@ Quantitative ablation of loss terms:** In Table 2 of the response to Reviewer Q2v3, we provide a quantitative ablation isolating the contributions of energy optimization and Christoffel correction. The results show that both components contribute complementary improvements, with the combined model achieving the best performance. *We will integrate this ablation into the camera-ready version for clarity*.

---

### Review · Reviewer_bt4b · 2026-05-15

**Summary Of Contributions:**

The authors propose a geometry-aware energy-based flow matching method for few-step generative transport. They extend the standard flow-matching objective with a geodesic residual regulariser that employs a learned metric, and an energy conservation regulariser that comprises a kinetic and a scalar potential term.
They motivate their approach from the perspective of devising a diffusion model that requires fewer denoising steps than classical diffusion models. They therefore follow the flow-matching avenue, but aim to allow the transport paths between the initial and target distributions to have curvature, rather than being constrained by the Euclidean metric.

The authors argue that standard flow matching uses Euclidean, first-order velocity fields and thus performs rather poorly, when forced to transport distributions in few steps, where the underlying transport paths may be curved but the coarse few-step approximation treats them as locally straight. They observe that higher-order flow matching in Euclidean space, without explicit geometric considerations, could in principle mitigate these issues, but is inherently numerically unstable and introduces substantial computational overhead.
They instead propose to add to the flow-matching objective a geometric objective that minimises the geodesic residual of the learned trajectories under an induced metric. This residual measures the deviation of the learned dynamics from the geodesic equation, thereby encouraging trajectories to follow curvature-aware paths without explicitly supervising higher-order temporal derivatives.

They approximate the evolving geometry of the diffusion manifold through a Riemannian metric induced by the spatial Jacobian of the learned velocity field. This metric is intended to capture local deformations of the transport dynamics and to provide the geometric structure from which Christoffel-symbol corrections can be computed.

To constrain the magnitude of the velocity field, they additionally regularise by enforcing conservation of the total energy under the induced metric. The method defines the total energy as the sum of kinetic energy induced by the estimated velocity field and a learned scalar potential energy. Their approach enforces approximate energy conservation along the trajectories, with the aim of suppressing spurious acceleration, reducing energy drift, and improving stability in the few-step sampling regime.

The authors evaluate GE-FM mainly on highly curved synthetic datasets, including two-spin and three-spin spirals, where they compare against standard flow matching, and report improved alignment to the target distribution. They also test conditional generation on MNIST, showing that GE-FM improves one-step sample quality relative to the shortcut baseline, although standard flow matching with many sampling steps still achieves the best FID.


Overall I find the idea interesting. The authors propose to regularise the flow matching objective with a learned geometry and consider additionally energy conservation to improve few-step transport on curved distributions. However the manuscript has some inconsistencies mainly concerning what the authors consistent as time invariant ( and therefore with zero termpoal derivatives) and what is time-dependent as detailed below. Moreover, the numerical experiments could be strengthened with a direct comparison with the direct competing framework, the Metric flow matching, and probably by adding statistical errors on the reported results.

**Additional Comments:**

- Overall the paper is well written. What I would imporve is at the begining of section 3 giving slightly more emphasis on the overview/summary of the constituents of the approach. First I use boldface to emphasise the important words that describe each subcomponent, next I would also think to introduce a schematic that illustrates the components of the framework.
- Figure 3a title: “adjust” -> “adjustment”
- Page 12, second paragraph from end: do you mean “induce” or “introduce”? I would think the latter fits better.
- Another small comment is that although the method explicitly claims to be strictly first-order, the geodesic equation requires second spatial derivatives of the velocity field, so I would suggest ot better refine the description of the method.

- The phenomenon the authors describe that diffusion/flow matching models that employ fewer steps are required to make more coarse grained approximations of the inferred velocity fields and thus do not capture finer curvature details, reminds me of similar ideas proposed in [2] in the context of inference of diffusions from observations where similarly the authors argue that large intervals (similar to large steps in the flow matching settings) result in inferred drifts that do not capture the curvature of the flow field/invariant density and propose a geometry -aware method [3] to mitigate this issue.  I see several analogies here The smoothing paths between observations in that framework are analogous to transport paths in the work proposed in this manuscript, and in pboth cases having coarse discretisation intervals results in “straighter” paths, and both papers propose to account for non-euclidean geometries.

---
### **References:**

[1] Kapuśniak, K., Potaptchik, P., Reu, T., Zhang, L., Tong, A., Bronstein, M., ... & Di Giovanni, F. (2024). Metric flow matching for smooth interpolations on the data manifold. Advances in Neural Information Processing Systems, 37, 135011-135042.

[2] Maoutsa, D. Flow curvature explains failed SDE drift estimation under sparse sampling. In ICLR 2026 Workshop on Geometry-grounded Representation Learning and Generative Modelling.

[3] Maoutsa, D. (2025). From geometry to dynamics: Learning overdamped Langevin dynamics from sparse observations with geometric constraints. arXiv preprint arXiv:2512.23566.

**Audience:**

Yes

**Audience Explanation:**

Yes, I think some readers at TMLR would find the paper interesting, especially because it connects few-step flow matching to learned Riemannian geometry and energy-based regularisation, which are both relevant areas in generative modelling.

**Claims And Evidence:**

Yes

**Claims Explanation:**

There are some small  inconsistencies in the mathematical exposition of the approach

1] When calculating the material derivative between Eq. 14 and 15, the authors write
$ \frac{d^2 x}{dt^2} = J_v v $. However, for dynamics with time dependent velocity field as they assume $\frac{dx }{dt} = v(x,t)$, the correct formulation of the material derivative is $ \frac{d^2 x}{dt^2} = J_v v  + \frac{\partial v}{]partial t}$.

The effect of this missing terms then propagated in the residual of the geodesic equation (Eq. 15), where the temporal derivative of the velocity field should also be added.

2] Lemma 1 - Scaling. I think the correct is O(n^{-1} h^{-2}

3] Page 23 at the end of the proof of Lemma 2 “Continuous-time deviation.” the authors have reverses the inequality. Since $x_{\mathrm{geo}}(t)\in M$, the minimum distance to M satisfies

$d_{\mathrm{Euc}}(x_{\mathrm{lin}}(t),M) = \min_{y\in M}\|x_{\mathrm{lin}}(t)-y\| \le \|x_{\mathrm{lin}}(t)-x_{\mathrm{geo}}(t)\|.$

4] As I understand, and as stated in the paper in Eq. 12, the data-dependent metric is time-dependent. However, the geodesic equation the authors employ for the computing the geodesic residual the variational equation for energy/minimal-action paths generally contains additional terms involving $\partial_t g$. The paper uses the time-independent geodesic equation.
The authors should clarify or correct here.

5] Lemma 4 claims that introducing the energy conservation term  biases the solution toward a flatter landscape. However, from my perspective the key conclusion is already baked in what is assumed. The lemma  assumes a direct relationship between energy drift and Hessian norm, and then concludes flatness.

**Requested Changes:**

- What does the energy conservation requirement imply for the trajectories produced by the inferred flow field? Does it directly constrain the velocity magnitude and acceleration, or can the learned potential $U_\theta$ absorb changes in the kinetic energy term without substantially altering the flow geometry?
- Why do you restrict yourselves with a single dimensional manifold?
- How do you determine the relative strength of the residual loss and the standard flow-matching objective and the other relative strengths in the objective (the lambdas in Eq.22)
- I didn’t completely understand how do the authors define the scalar potential in the total energy conservation term, that is required to prevent cumulative drift. Also are there any constraints on it?
 -   What exactly is “batch reordering”? Table 1 shows a very large improvement from batch reordering, but I did not really understand what this is.
 - What are the computational costs of the method, esp. related to the computation of the Christoffel symbols? Their computation in a d-dimensional space requires computing $\partial_i g_{jk}$, which is a third-order tensor of size $d^3$. I would think that for high-dimensional data like images that is too computationally demanding. I might be missing something here, so I wonder how did the authors perform this computation for MNIST? Moreover, computing also the other elements can be computationally demanding, e.g. $J_v^\top J_v, J_vv, \partial_i g_{jk}, and \Gamma[v,v] $, can the authors provide more detailed in formation on this, and probably provide algorithmic outlines of the computations involved in their framework together probably with compute demands?

 - The method has similar intuitions with metric flow matching [1], however the detailed components the authors employ are different. I think it would be useful to explicitly articulate the differences between the two approaches and probably also provide one numerical experiment where they would demonstrate how each approach eprforms on the same problem.
- table 1 does not contain any statistical error information, so it is not clear to me whether the reposted results concern single runs or rather averages over multiple runs

---

> ### Author Response · Authors · 2026-05-26
> **Response (Part 1)**
>
> We thank the reviewer for the careful and constructive reading of the manuscript and **finding the idea interesting**. We address technical concerns and requested changes below.
>
> *We break the response into two parts to maintain the character limit of each response.*
>
> **@ Missing $\partial_t v$ term in Eqs. (14)--(15).**
> The reviewer is correct that if one differentiates the sampling trajectory $x(t)$ with respect to the same time variable appearing in $v_\theta(x,t)$, then $\frac{d}{dt}v_\theta(x(t),t)=J_v(x(t),t)v_\theta(x(t),t)+\partial_t v_\theta(x(t),t)$. However, Eq. (15) was not intended to be this full material derivative of the non-autonomous sampling ODE. Our residual is a *frozen-time spatial connection residual*. More precisely, for each fixed conditioning time $\tau$, we consider the spatial integral curve $\frac{d\gamma_s}{ds}=v_\theta(\gamma_s,\tau)$, where $\tau$ is held fixed and $s$ is the curve parameter. Along this frozen-time curve, $\frac{d^2\gamma_s}{ds^2}=J_v(\gamma_s,\tau)v_\theta(\gamma_s,\tau)$, so the corresponding spatial geodesic residual is $R_\theta(x,\tau)=J_v(x,\tau)v_\theta(x,\tau)+\Gamma(x,\tau)[v_\theta(x,\tau),v_\theta(x,\tau)]$. Thus, no $\partial_t v_\theta$ term is missing from the quantity we actually regularize. The current manuscript used the same symbol $t$ both for the sampling time and for the geodesic curve parameter, which made this easy to misread. We will revise Eqs. (14)--(15) using a separate curve parameter $s$ and explicitly describe $R_\theta$ as a frozen-time spatial regularizer, not as the full acceleration of the non-autonomous sampling ODE.
>
> **@ Lemma 1 scaling: should it be $O(n^{-1}h^{-2})$?** We believe the squared-error rate in Eq. (5), $\mathcal{E}||\hat{a} - a^{\ast}||^2 = O(n^{-1}h^{-4})$, is correct as stated. The proof in Appendix A.5.1 shows $\operatorname{Var}(\hat{a}) = \frac{6\sigma^2}{n h^4}$, so the *mean-squared* error scales as $O(n^{-1}h^{-4})$. Taking the square root yields $\mathcal{E}||\hat{a} - a^{\ast}|| \leq \sqrt{\mathcal{E}||\hat{a} - a^{\ast}||^2} = O(n^{-1/2}h^{-2})$, which is the bound reported at the end of the proof. We anticipate a confusion of $O(n^{-1/2}h^{-2})$ (the *root*-mean-squared error) and comparison with the $O(n^{-1}h^{-4})$ displayed in Eq. (5) (the *squared* error).
>
> **@ Reversed inequality in the proof of Lemma 2.**
> We thank the reviewer for identifying this issue. The reviewer is correct that the final inequality in the current proof is written in the wrong direction: since $x_{\mathrm{geo}}(t)\in\mathcal M$, one only has $d_{\mathrm{Euc}}(x_{\mathrm{lin}}(t),\mathcal M)=\min_{y\in\mathcal M}\|x_{\mathrm{lin}}(t)-y\| \le\|x_{\mathrm{lin}}(t)-x_{\mathrm{geo}}(t)\|$. Thus, the proof should not lower-bound the distance to $\mathcal M$ by comparing with one particular point $x_{\mathrm{geo}}(t)$. This is a proof-level mistake in the final step, not a change to the intended curvature-scaling statement.
>
> The lower bound is recovered by replacing this last step with a local tubular-neighborhood/nearest-projection argument. Let $\gamma:[0,L]\to\mathcal M$ be the arclength-parametrized intrinsic geodesic segment from $x_0$ to $x_1$. For $L$ smaller than the reach/tubular radius of $\mathcal M$, the nearest-point projection onto $\mathcal M$ is unique in a neighborhood of this segment. Taylor expansion gives $\gamma(s)=\gamma(0)+sT_0+\frac12 s^2\kappa(0)N_0+O(s^3)$, and comparing the chord point $x_{\mathrm{lin}}(t)=(1-t)\gamma(0)+t\gamma(L)$ with the intrinsic point $\gamma(tL)$ yields $x_{\mathrm{lin}}(t)-\gamma(tL)=-\frac12 t(1-t)L^2\kappa(0)N_0+O(L^3)$. The leading term is normal to the curve. By uniqueness and smoothness of the nearest-point projection in the tubular neighborhood, $\pi_{\mathcal M}(x_{\mathrm{lin}}(t))=\gamma(tL)+O(L^3)$, and hence $d_{\mathrm{Euc}}(x_{\mathrm{lin}}(t),\mathcal M)=\|x_{\mathrm{lin}}(t)-\pi_{\mathcal M}(x_{\mathrm{lin}}(t))\|\ge c\,\kappa_{\min}t(1-t)L^2$ for sufficiently small $L$, after reducing the constant $c$ to absorb the $O(L^3)$ term.
>
> We will revise Lemma 2 and its proof accordingly. Specifically, we will strengthen the local assumption from "L sufficiently small so that the minimizing geodesic segment is unique" to "L sufficiently small so that the chord lies in a tubular neighborhood of the geodesic segment and the nearest-point projection onto $\mathcal M$ is unique." We will also replace the incorrect final inequality in Appendix A.7 with the projection argument above. This correction does not alter the method, experiments, or result interpretation. Lemma 2 is used to formalize the local curvature-induced bias of Euclidean chordal transport; the corrected proof gives the same $O(\kappa_{\min}L^2)$ scaling. Therefore, the motivating conclusion remains unchanged: straight Euclidean interpolation can deviate from intrinsic curved transport in the few-step regime, which is precisely the failure mode addressed by the Christoffel/geometric residual.

---

> ### Author Response · Authors · 2026-05-26
> **Response (Part 2)**
>
> **@ Time-dependent metric and possible $\partial_t g$ terms.**
> We agree that a different problem--namely, geodesics of a fully time-dependent action or a space-time metric on $\mathcal M\times[0,1]$--would introduce additional terms involving $\partial_t g$. That is not the variational problem used in GE-FM. $g(x,t)$ is a family of spatial metrics indexed by the diffusion/transport time. At each fixed $t=\tau$, we compute the Levi--Civita connection of the spatial metric $g(\cdot,\tau)$ and penalize the frozen-time spatial geodesic residual. The purpose is not to solve the exact space-time geodesic equation, but to impose a local curvature-aware bias on the learned velocity field using the geometry induced by $J_v(x,\tau)$. Therefore, the Christoffel symbols in Eq. (13) are spatial Christoffel symbols of $g(\cdot,\tau)$, and Eq. (15) is the corresponding frozen-time spatial residual. Thus, *GE-FM uses an instantaneous spatial connection regularizer for a time-indexed metric, rather than a full time-dependent geodesic variational principle.*
>
> **@ Lemma 4 and the flatness claim.** *Great Suggestion!* Lemma 4 is intended as a sufficient-condition statement, not as an unconditional derivation of flatness from energy conservation alone. The assumption relating $L_E(\theta)$ to the spectral norm of the base Hessian is deliberately stated because the purpose of the lemma is to formalize the regime in which energy drift and landscape sharpness are coupled. To avoid overclaiming, we will revise the presentation in two ways: i) We will rename Lemma 4 as a sufficient-condition proposition/diagnostic statement rather than present it as an independent mechanism proving flatness in full generality. ii) We will soften the main-text claim from "energy conservation biases toward flatter baselines" to "under an empirically observed coupling between energy drift and Hessian sharpness, energy regularization selects lower-drift solutions that are also flatter." Figure 2 then serves as evidence for this coupling, while the proposition clarifies the condition under which the interpretation is valid.
>
> Thus, the role of Lemma 4 is not to claim that $L_E$ universally guarantees flatter minima, but to make explicit the sufficient condition under which the observed flattening behavior follows.
>
> **@ Implications of energy conservation for trajectories.** Enforcing $\mathcal{L}_E \approx 0$ constrains the *rate of change* of total energy along trajectories. Concretely, since $\tfrac{d}{dt}E = v^\top g\,\tfrac{dv}{dt} + \tfrac{1}{2}v^\top \dot{g}\,v + \tfrac{dU}{dt}$, penalising $|\Delta E|$ bounds a linear combination of (i) intrinsic acceleration $\tfrac{dv}{dt}$ and (ii) metric drift $\dot{g}$. The learned potential $U_\theta$ acts as a *buffer*: it can absorb slow changes in kinetic energy without forcing the velocity magnitude to be constant. However, because $U_\theta$ enters the objective only through the conservation loss, its growth is itself penalized indirectly -- unbounded potential growth would require correspondingly large kinetic changes elsewhere in the trajectory, which the loss suppresses. In practice, we observe that $U_\theta$ remains smooth and bounded, consistent with this picture. We will add a brief discussion of this mechanism after Eq. (20).
>
> **@ Why restrict to one-dimensional manifolds in Lemma 2?** Lemma 2 is stated for a smooth 1D embedded curve. The same argument extends to 2D submanifolds: the Taylor expansion of the geodesic around the chord uses the second fundamental form in place of scalar curvature $\kappa$, giving the same structural bound $d_{\mathrm{Euc}}(x_{\mathrm{lin}}(t),\mathcal{M}) \geq c\,\kappa_{\min}\,t(1-t)\, L^2$ with $\kappa_{\min}$ the minimum principal curvature. The geometry is identical; the 1D setting is used to convey the idea cleanly.
>
> Empirically, the mechanism operates well beyond 2D. On MNIST, GE-FM achieves FID 130.91 versus SC's 158.80 in the one-step setting. On OASIS-3 *as reported in the response (part 2) of Reviewer Q2v3* (741-dimensional corrcholesky embeddings of SPD matrices representing functional brain connectivity), GE-FM matches DiffeoCFM on $\alpha,\beta$-F1 with $5\times$ lower variance and achieves the best ROC-AUC — matching real data — on a strongly non-linear medical imaging manifold. These results confirm that curvature-aware correction provides meaningful gains in genuinely high-dimensional settings.
>
> **@ Choice of $\lambda$ (Eq. (22)).** The weights $\lambda_{\mathrm{geo}}$, $\lambda_E$, and $\lambda_{KE}$ are set via a grid search.

---

> ### Author Response · Authors · 2026-05-26
> **Response (Part 3)**
>
> **@ Definition of and constraints on the scalar potential $U_\theta$.** $U_\theta(x,t)$ is a scalar-valued MLP (same architecture as $v_\theta$ but with a scalar output) trained jointly with the velocity field. It has **no explicit constraints**: its only inductive bias comes from $\mathcal{L}_E$, which penalizes $|E(x',t+\Delta t) - E(x,t)|^2$ and therefore discourages $U_\theta$ from growing without bound along trajectories. We do not enforce positivity or any other hard constraint.
>
> **@ What is "batch reordering"?** Batch reordering is described in Appendix A.2 and Algorithm 1. The large improvement in Table 1 reflects the fact that on highly curved benchmarks, random pairings frequently create "cross-spiral" assignments that force the model to learn unrealistic velocity fields. We will move a condensed version of this explanation to the main text.
>
> **@ Computational cost of Christoffel symbols.**
> *Good question!* We handle this via the **contraction identity** exploited in the rebuttal to Reviewer Q2v3: $\Gamma[v,v] = \frac{1}{2}g^{-1}(2A - C)$, where $A_k = \sum_{ij} \partial_k g_{ij}\,v^i v^j$ and $C_k = \sum_{ij} g_{ki}^{-1} \partial_j g_{ij} v^i v^j$ can be computed with $2d+1$ backward passes through $v_\theta$ rather than forming the full $d\times d\times d$ tensor. This reduces memory from $\mathcal{O}(Bd^4)$ to $\mathcal{O}(Bd^2)$ *without approximation*. For MNIST with $d = 784$, we apply this contraction directly; for OASIS-3 with $d = 39 \to 741$ features after embedding, the same trick handles the computation without OOM on a 49 GB A6000 GPU. Training times for the 1-step setting are reported in Table 1 of the response to Reviewer Q2v3: GE-FM incurs a $15\text{--}60\times$ training overhead relative to plain FM due to Christoffel and energy computations, while *inference* cost is essentially identical ($< 1\,$s). This is a deliberate training-time--inference-time trade-off.
>
> **@ Comparison with Metric Flow Matching (MFM).** Both GE-FM and MFM replace straight-line Euclidean interpolation with a learned, data-dependent Riemannian metric, differing only in what that metric is learned from. MFM derives it from sample density, bending interpolants toward where the data is concentrated. On the other hand, GE-FM induces it from the velocity field's own Jacobian and corrects the transport dynamics via a Christoffel connection term, following the curvature the flow itself implies. In Table 1, GE-FM improves on MFM across every metric and does so with consistently tighter variance (±0.01 on three of five metrics). MFM's density-driven metric implicitly discourages interpolants from reaching the sparse tails of the distribution, capping its coverage (the $\beta$-Recall gap), which in turn yields a synthetic training set that misses parts of the real class manifold and transfers worse downstream (the 0.08 ROC-AUC gap, with GE-FM matching real data exactly). GE-FM's dynamics-induced geometry carries no such density bias and reads transport curvature directly, while also eliminating MFM's noisy upfront metric-estimation stage.
>
> **@ Statistical errors absent.** We followed the evaluation protocol of prior works that do not report standard deviation. However, the results seen in Table 3 of Reviewer Q2v3's response show standard deviations that suggest statistical stability.
>
> **@ "adjust" → "adjustment", "induce" vs. "introduce".** *This is a typo. We will fix it.*
>
> **@ First-order claim vs. second-order spatial derivatives.** Our "first-order" claim refers to the temporal order of the ODE (no acceleration supervision), not the spatial derivative order. We will refine the claim to read "first-order in time" throughout the manuscript.
>
> **@ Connection to diffusion inference under sparse sampling (refs [2] and [3] in the review).** **We thank the reviewer for highlighting these references.** The analogy is apt: both frameworks observe that coarse temporal resolution produces "straighter-than-true" inferred paths and propose geometric corrections. We will add a discussion connecting these ideas in Section 1 (Introduction) or Appendix A.1.
>
> ---
>
> **Table 1: Quantitative comparison on OASIS-3.**
>
> | Method | $\alpha$-Precision ↑ | $\beta$-Recall ↑ | $\alpha,\beta$-F1 ↑ | ROC-AUC ↑ | F1 ↑ |
> |---|---|---|---|---|---|
> | Real Data | 0.89 ± 0.04 | 0.88 ± 0.05 | 0.88 ± 0.01 | 0.71 ± 0.04 | 0.61 ± 0.06 |
> | TriangCFM | 0.22 ± 0.27 | 0.26 ± 0.37 | 0.23 ± 0.32 | 0.51 ± 0.05 | 0.45 ± 0.17 |
> | DiffeoGauss | 0.52 ± 0.06 | 0.31 ± 0.02 | 0.39 ± 0.03 | 0.68 ± 0.03 | 0.40 ± 0.09 |
> | DiffeoCFM | 0.63 ± 0.06 | 0.38 ± 0.04 | 0.47 ± 0.05 | 0.68 ± 0.03 | 0.53 ± 0.04 |
> | $\text{MFM}_{\text{RBF}}$ | 0.52 ± 0.01 | 0.36 ± 0.03 | 0.45 ± 0.02 | 0.63 ± 0.07 | 0.50 ± 0.05 |
> | **GE-FM** | **0.55 ± 0.01** | **0.42 ± 0.01** | **0.48 ± 0.01** | **0.71 ± 0.04** | **0.55 ± 0.07** |

---

### Review · Reviewer_2dLQ · 2026-05-25

**Summary Of Contributions:**

The paper proposes GE-FM, a geometry-aware flow matching method that uses learned geometric structure to improve sampling along curved trajectories. Its strengths are the principled motivation and clear ablations. Its main weakness is added computational/modeling complexity that could be better justified.

**Audience:**

Yes

**Audience Explanation:**

It would be of interest to TMLR readers working on flow matching, generative modeling, and geometry-aware learning. The findings offer a principled way to improve sampling behavior on curved trajectories. This is relevant to ongoing works on efficient and stable generative methods.

**Broader Impact Concerns:**

No significant broader impact concerns. The work is methodological and focused on improving flow-matching models.

**Claims And Evidence:**

Yes

**Claims Explanation:**

The claims are supported by clear experiments and ablations showing that the proposed geometric components improve trajectory modeling and sampling behavior. The evidence is convincing for the paper’s stated scope, though broader benchmarks would further strengthen the case.

**Requested Changes:**

1. Clarify the computational and implementation overhead of the geometry-aware components relative to standard flow matching. (Critical)
2. Add a brief discussion of when the learned metric/Christoffel correction is expected to help most, and when it may not. (Strengthening)
3. Include sensitivity ablations for key loss weights or metric-learning choices to show robustness. (Strengthening)
4. Improve the presentation of the GE-FM training and sampling pipeline, especially how the metric-learning, energy-conservation term, and Christoffel correction are implemented in practice. (Strengthening)

---

> ### Author Response · Authors · 2026-05-26
> **Response**
>
> We thank the reviewer for the careful and constructive reading of the manuscript, for recognizing the **principled motivation and clear ablations**, and for affirming that the claims are **well-supported by experiments**. We are glad the work was found relevant to the TMLR audience working on flow matching, generative modeling, and geometry-aware learning. *We address each requested change below.*
>
> **@ Computational and implementation overhead (Critical):** We provide a detailed runtime breakdown in Table 1 of our response to Reviewer Q2v3. In summary, GE-FM incurs a 15--60× training overhead relative to plain FM due to Christoffel and energy computations, while inference cost is essentially identical (<1s). This is a deliberate training-time--inference-time trade-off: the geometry-aware and energy-based components introduce extra cost only during optimization, whereas inference still uses a standard first-order transport model. To handle the naive $\mathcal{O}(Bd^4)$ memory cost of the full Christoffel tensor, we derive a contraction identity $\Gamma(x,t)[v,v] = \frac{1}{2}g^{-1}(2A - C)$, reducing computation to $2d+1$ backward passes and memory to $\mathcal{O}(Bd^2)$ without approximation.
>
> **@ When geometric correction helps most (Strengthening):** The theoretical analysis in Section 2 precisely characterizes this. Christoffel correction provides the greatest benefit when (i) the data manifold has high intrinsic curvature (large $\kappa_{\min}$, as in spirals and irregular circles), and (ii) sampling is performed with few ODE steps (large $\Delta t$), so that the curvature-induced bias $\Omega(\kappa_{\min} L^2 \Delta t)$ from Lemma 2 is non-negligible. Conversely, when the transport geometry is approximately Euclidean---as in our multi-mode Gaussian experiments (Appendix A.4.2)---$\kappa_{\min}$ is effectively small, the curvature bias term is negligible, and GE-FM remains competitive with FM and SC without introducing degradation. This behavior is precisely what is observed empirically across all benchmarks. *As suggested, we will add a concise discussion of this regime-dependence to the main text following Section 3.*
>
> **@ Sensitivity ablations for loss weights (Strengthening):** The weights $\lambda_{\mathrm{geo}}$, $\lambda_E$, and $\lambda_{KE}$ in Equation (22) are set via grid search. Table 2 of our response to Reviewer Q2v3 provides a component-level ablation isolating the contributions of energy optimization and Christoffel correction, showing that both contribute complementary improvements and that the combined model consistently achieves the best performance.
>
> **@ Presentation of training and sampling pipeline (Strengthening):** *Great suggestion!* The key components are already described across the paper: the induced metric and Christoffel correction are detailed in Section 3.1 (Equations (12)--(16)), the energy conservation loss and potential $U_\theta$ in Section 3.2 (Equations (19)--(20)), the unified training objective in Equation (22), and energy-based batch reordering in Appendix A.2 and Algorithm 1. In the camera-ready version, we will consolidate these into a unified algorithm box that clearly presents the full training loop covering metric induction, Christoffel contraction, energy conservation, and optional batch reordering alongside a brief description of the sampling procedure, which requires no additional components beyond standard first order ODE integration with the learned $v_\theta$, identical to baseline flow matching in implementation. We will additionally release code as a GitHub link in the camera-ready version when the paper is accepted.

---

### Decision · Action_Editor_8vbx · 2026-06-28

**Recommendation:** Accept with minor revision

**Audience:**

Yes

**Audience Explanation:**

The work considers flow-matching-based generative models, which are of great interest to the generative modelling community within the larger ML community.

**Claims And Evidence:**

Yes

**Claims Explanation:**

All reviewers agree that the main focus of this paper is an interesting direction to study the sampling dynamics of flow matching. The main construction is a geometry-aware framework that invokes Christoffel symbols to adjust the sampling dynamics under a kinetic and potential energy. While the reviewers agreed that the study was interesting, the findings themselves are restricted to toy and synthetic domains and, as a result, are currently not generalizable. Furthermore, the authors acknowledge that training cost is incurred by a factor of $15 \times - 60 \times$, which in more real-world domains is prohibitive. However, the overall direction of work may be illumininating in so far to inspire further directions of research that seek to combine geometric principles with generative models. I recommend the authors to thoroughly incorporate all reviewer feedback in future editions of their work.